# The colonization of the oceans by calcifying pelagic algae

Baptiste Suchéras-Marx[1]*, Emanuela Mattioli[2,3]*, Pascal Allemand[2], Fabienne Giraud[4], Bernard Pittet[2], Julien Plancq[5], Gilles Escarguel[6]

[1]Aix Marseille Univ, CNRS, IRD, INRA, Coll France, CEREGE, Aix-en-Provence, France.
[2]Université de Lyon, UCBL, ENSL, CNRS, LGL-TPE, F-69622 Villeurbanne, France.
[3]Institut Universitaire de France, Paris, France.
[4]Univ. Grenoble Alpes, Univ. Savoie Mont Blanc, CNRS, IRD, IFSTTAR, ISTerre, 38000 Grenoble, France.
[5]School of Geographical and Earth Sciences, University of Glasgow, Glasgow G12 8QQ, UK.
[6]Université de Lyon, UMR 5023 LEHNA, UCBL, CNRS, ENTPE, F-69622 Villeurbanne, France
*Both authors contributed equally to this work

*Correspondence to*: Baptiste Suchéras-Marx (sucheras@cerege.fr); Emanuela Mattioli (emanuela.mattioli@univ-lyon1.fr)

**Abstract.** The rise of calcareous nannoplankton in Mesozoic oceans has deeply impacted ocean chemistry and contributed to shape modern oceans. Nevertheless, the calcareous nannoplankton colonization of past marine environments remains poorly understood. Based on an extensive compilation of published and unpublished data, we show that their accumulation rates in sediments increased from the Early Jurassic (~200 Ma) to the Early Cretaceous (~120 Ma), although these algae diversified up to the end of the Mesozoic (66 Ma). After the middle Eocene (~45 Ma), a decoupling occurred between accumulation rates, diversity and coccolith size. The time series analysed points toward a three-phase evolutionary dynamic. An Invasion phase of the open-ocean realms was followed by a Specialization phase occurring along with taxonomic diversification, ended by an Establishment phase where few small-sized species dominated. The current hegemony of calcareous nannoplankton in the World Ocean results from a long-term and complex evolutionary history shaped by ecological interactions and abiotic forcing.

## 1. Introduction

Calcifying pelagic algae, also known as calcareous nannoplankton, are an important and globally distributed component of marine biota both in terms of abundance and diversity. Calcareous nannoplankton is today mainly composed of coccolithophores, which are unicellular Haptophyta algae producing microscopic (1-20 µm) calcite platelets, the coccoliths, and occurring in the fossil record since the Late Triassic (~210 Ma; Gardin et al., 2012). Coccoliths, together with *incertae sedis* calcite remains, are grouped into calcareous nannofossils, and are abundantly recovered in Mesozoic and Cenozoic marine sediments. Coccoliths are produced inside the coccolithophore cell and are then extruded to form an extracellular, mineralised coccosphere. Although this calcification process requires energy from the cell, the reason why coccolithophores produce coccoliths remain uncertain (Monteiro et al., 2016). In modern surface oceans, coccolithophores perform ~1-10 % of the total organic carbon fixation, featuring in some cases more than 50 %, while calcification of coccolithophores contributes ~1-10 % of the total carbon fixation (Poulton et al., 2007). Nevertheless, their contribution to the carbon flux toward the ocean-

interior is twofold, since calcite also acts as ballast for the organic carbon (Klaas et al., 2002). Eventually, calcareous nannofossils represent about half of the extant pelagic carbonate sediments in the oceanic realm (Baumann et al., 2004; Broecker and Clark, 2009), and accounted even more in Neogene sediments despite their small size (Suchéras-Marx and Henderiks, 2014). Conversely, during the early coccolithophore evolution, they only represented a minor contribution to the

total calcium carbonate in sediments, with extremely low nannofossil accumulation rates in the Jurassic Period (Mattioli et al., 2009; Suchéras-Marx et al., 2012). There is then a transition from Jurassic calcareous nannofossil-poor to Late Cretaceous and Cenozoic calcareous nannofossil-rich oceanic sediments which has shifted the carbonate accumulation sustained by benthic organisms from neritic environments to an accumulation in pelagic environments supported by planktic organisms. This major carbonate system change is known as the *Kuenen* Event (Roth, 1989), and has been referred to a tectonically-mediated

intensification of the ocean circulation. This event is concomitant with the development of several planktic groups (e.g., planktic foraminifera (Hart et al., 2003), diatoms (Kooistra et al., 2007)), may be seen as a Mesozoic Plankton Revolution (derived from Vermeij, 1977) and thus is also dramatically related to plankton evolution. The causes and consequences of this biotic revolution have been extensively discussed, but the transition itself remains poorly documented; most interpretations solely rely on species richness (Falkowski et al., 2004; Knoll and Follows, 2016), which does not provide an exhaustive

framework to fully appreciate the evolutionary history of calcareous nannoplankton.

Our working hypothesis is that this difference between Jurassic and Cenozoic pelagic carbonate accumulation rates points toward a major change in the nannoplankton evolutionary dynamics through geological time, rather than being merely due to environmental changes. In order to test this hypothesis, we analysed in this study the ~200 Myr-long evolutionary history of calcareous nannoplankton based on an extensive compilation of both published and unpublished nannofossil accumulation

rates (NAR; Table S1, Fig. 1), species richness (Bown, 2005), and coccolith mean size (i.e., at the assemblage level; Aubry et al., 2005; Herrmann and Thierstein, 2012). The novelty of this study stands in the long-term reconstruction of NAR and its use as a proxy for assessing the evolutionary dynamics of the calcareous nannoplankton. Fossil-based quantification in the sedimentary record is most often overlooked in paleontological studies due to the uneven character of the fossil record, but the continuous and abundant record of calcareous nannofossils and their taphonomic resilience compensate most preservation and

sampling issues. Our approach therefore represents an unprecedented advance in understanding the evolutionary dynamics of a major planktic group. We discuss the resulting pattern with respect to the microplankton evolutionary history and compare it with the long-term global climate, oceanographic and environmental changes known for this time interval.

## 2. Materials and Methods

### 2.1 Sample preparation of the compiled data

All the published and unpublished data of nannofossil absolute abundance (see supplementary information (SI)) coming from samples analysed by the authors result from the preparation technique described by Geisen et al. (1999). All the published data from the literature compiled (except one) also used the same preparation technique. The preparation consists of a settling

method, where a known quantity (m; 10-30 mg) of homogeneous rock powder is diluted in water and let settle in the random settling device for 24 h on a cover slide situated at a depth of 2 cm (h) within the random settling device. Water is eventually evacuated from the settling boxes very slowly in order to avoid turbulence and powder remobilization. Finally, cover slides are mounted on microscope slides using Rhodopas B (polyvinyl acetate) and studied under a light polarized (linear) microscope

with ×1000 magnification. Usually a minimum of 300 nannofossils per sample are counted (n) or a minimum of 50 fields of view (fov) is observed, depending on the concentration of particles on the cover slide. The nannofossil absolute abundance is then calculated based on the Eq. (1):

$$X = \frac{(n \times v)}{(m \times fov \times a \times h)} \tag{1}$$

Where

X is the nannofossil absolute abundance (nannofossil/$g_{bulk}$)

n is the number of nannofossils counted

v is the volume of water in the device

m is the mass of sediment in suspension (g)

fov is the number of fields of view observed

a is the surface area of one field of view (cm²)

h is the height of water column above the cover slide (cm)

The only study not using the random settling preparation technique deals with the Polaveno section (Italy; Late Berriasian–Early Hauterivian) (Erba and Tremolada, 2004), where nannofossils were quantified in thin sections thinned to an average thickness of 7 µm. Absolute abundances were then obtained by counting all nannofossil specimens on 1 mm² of the thin section

in a light polarized microscope with ×1250 magnification.

## 2.2 Accumulation rate calculation

The nannofossil accumulation rate is calculated using sedimentation rate following the Eq. (2):

$$NAR = NannoAb \times SR \times DBD \tag{2}$$


Where

NAR is nannofossil accumulation rate (nannofossil/m²/yr)

NannoAb is the nannofossil absolute abundance (nannofossil/$g_{bulk}$)

SR is the sedimentation rate (m/Myr)

DBD is the dry bulk density of the rock (g/cm³)

Sedimentation rates have been calculated based on the International Chronostratigraphic Chart 2012 (Gradstein et al., 2012). When cyclostratigraphy was available, we used the cycles provided by authors after re-evaluation of at least one anchor age (commonly a stage limit or a biostratigraphic datum). When cyclostratigraphy was not available, we used anchored ages mostly based on biostratigraphic datums, assuming that the sedimentation rate was constant between two datums. The dry bulk density

of rocks is missing in all but one Mesozoic studied samples (Suchéras-Marx et al., 2012). A typical value at 2.7 $g/cm^3$ corresponding to the calcite density was set when density was missing; this value is close to the 2.55 $g/cm^3$ measured for Middle Jurassic rocks (Suchéras-Marx et al., 2012), leading to a negligible difference in nannofossil accumulation rates.

For the Polaveno section samples, the calcareous nannofossil accumulation rates were calculated by the authors per unit area (1 $mm^2$) and time (1 yr) (Erba et al., 2004). The latter was derived from sedimentation rates estimated for individual magnetic

polarity chrons (Channel et al., 2000).

## 2.3 Data set compilation

All data source but one used the same preparation technique (see details above), limiting the discrepancies due to methodological differences. All the sites considered for nannofossil accumulation rate compilation are presented on a map (Fig. 1). The vast majority of the samples are from the Northern Hemisphere, and almost all samples for Jurassic and

Cretaceous times are from Western Europe outcrops – a relatively poor quantitative record of nannofossils exists outside Europe and in oceanic sites issued from deep-sea drilling programs. Europe, North Sea, Greenland and North Atlantic represent 81.16% of all compiled samples. Thus, results based on NAR, particularly for the Mesozoic which represent 84.31% of all sample compiled, will be mostly based on European/Atlantic localities and thus may describe pattern that occurred mainly in the Western Tethys and North Atlantic (see SI S3 and Fig. S3). For the Cenozoic, the data are more widely distributed but the

number of sample per Myr are less abundant than in the Mesozoic (see SI S3 and Fig. S3). All data compiled are provided in an Excel file, with one sheet per site or manuscript (Table S1) for a total of 3895 data points across 79 sites or manuscripts. Name, location and associated references for each site are provided in SI.

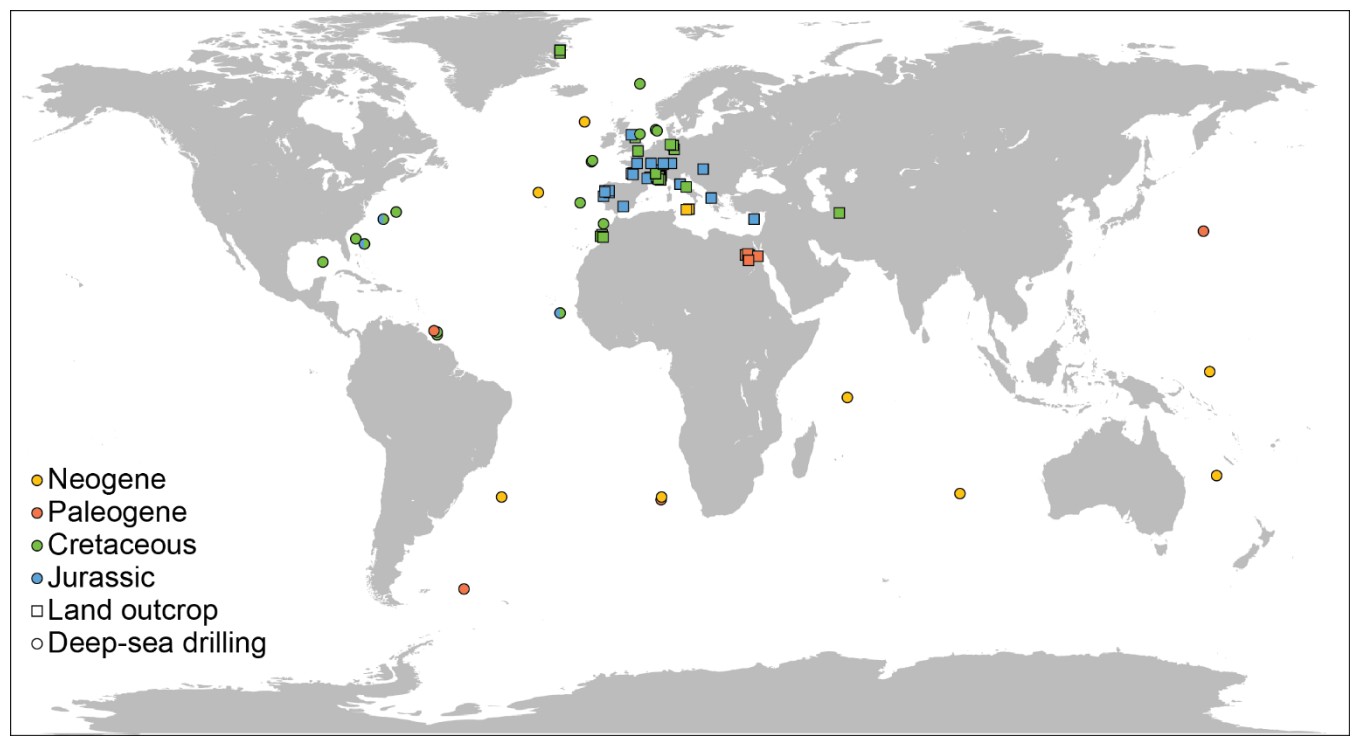

**Figure 1: Location of the sites compiled for this study. Colors indicate the age of the samples (Neogene, yellow; Paleogene, orange; Cretaceous, green; Jurassic, blue). Squares represent land outcrops and circles represent deep-sea drilling.**

### 2.4 Trend smoothing

For nannofossil accumulation rates and $pCO_2$ values (ppm and µatm; Foster et al., 2017, Witkowski et al., 2018) (Fig. 2; Figs. S1-S2; Fig. S4), a LOESS smoothed curve was computed in order to capture long-term variations and overlook short-term shifts that are more likely controlled by the number of studied sites, sampling resolution, and nannofossil preservation or, alternatively, by local environmental conditions. The data from Mejía et al. (2017) are excluded from the LOESS calculation due to the large uncertainties presented by the authors. See the SI for a discussion of the effect of the selected smoothing factor on the inferred trend. The curve was calculated using PAST3.24 (Hammer et al., 2001). The $CO_2$ curve was calculated using a smoothing factor of 0.1, and the nannofossil accumulation rate curve using a smoothing factor of 0.5, both associated with a 95 % bootstrapped Confidence Interval based on 999 random replicates.

### 2.5 Nannofossil accumulation rate paleomaps construction

Maps of nannofossil accumulation rates (Fig. 3, dataset in Table S2) have been drawn from the linear interpolation of the measurements performed in various sites using the dedicated matlab functions. The geographical coordinates of the sites studied were first converted in a sinusoidal projection that preserves distance ratios. The maps were then projected in a conformal Mercator projection in order to be more easily readable. The distance from continental coasts and the existence of

islands in the area of interpolation were not taken into account. We used the hypothesis that the islands were small enough for not spatially impacting the calcareous nannofossil accumulation rates. Continental coastlines were not used as a limit in the interpolation because they would have generated artificial variations due to the relatively high average distance between sites.

## 3. Results

Nannofossil accumulation rate (NAR), expressed as number of specimens per m² and per year, strongly varies between sites, but also stratigraphically within a single site (Fig. 2). In this study, we used a LOESS smoothing to catch the long-term trend and overlook short-term variations that may be influenced by preservation or local environmental conditions (Fig. 2, Fig. S1). Clearly, the resulting time series of smoothed NAR shows two main successive intervals: (i) a two-order of magnitude increase during the Jurassic and Early Cretaceous (i.e., from ~200 Ma to ~120 Ma), followed by (ii) a steady-state dynamic equilibrium up to end-Cenozoic times.

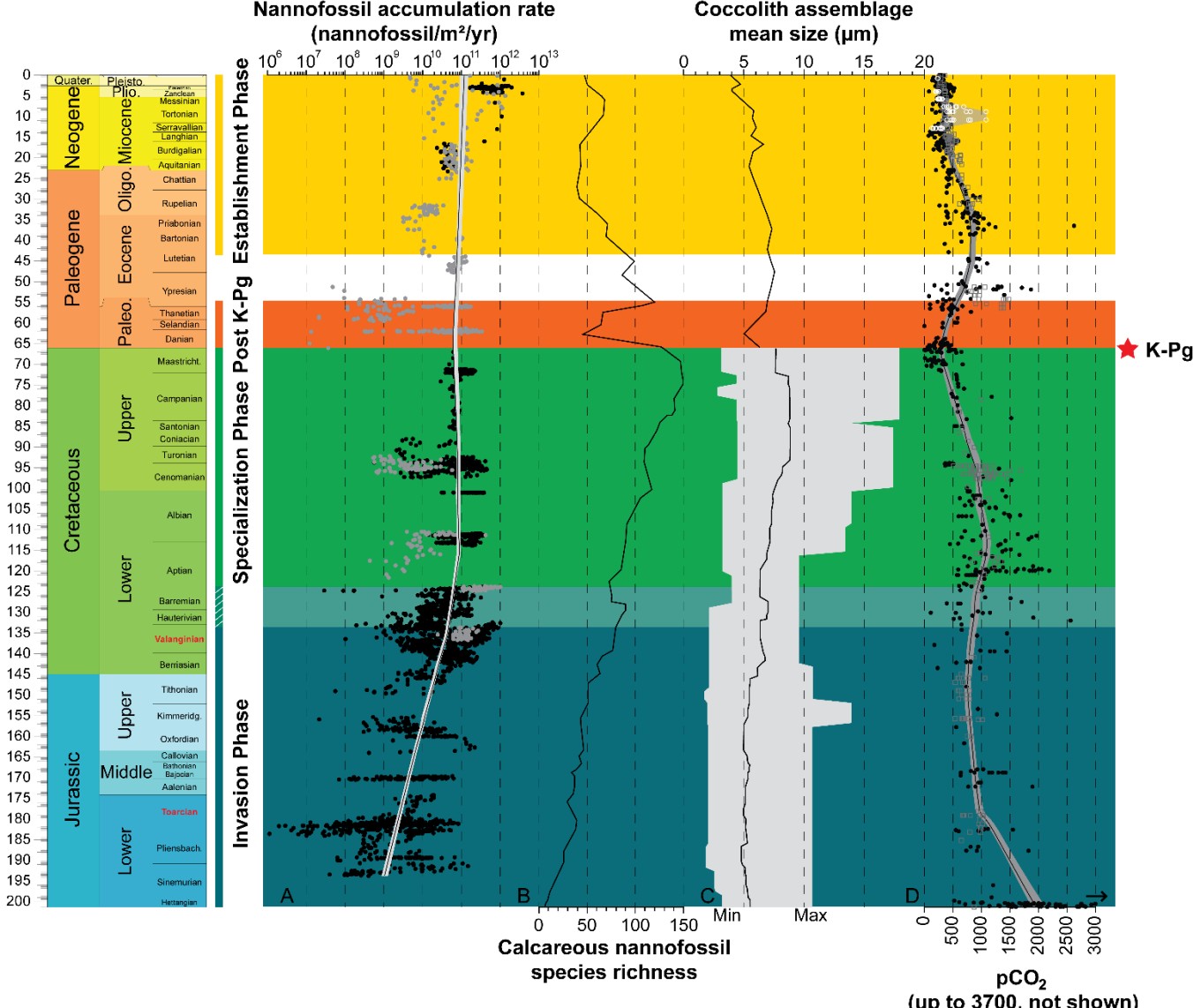

**Figure 2: Evolution through time of nannofossil accumulation rate, species richness and size. (A)** Compiled nannofossil accumulation rate (nannofossil/m²/yr). Black circles are samples from Europe, North Sea, Greenland and North Atlantic and grey circles from the rest of the world. The LOESS trend (SF 0.5) was calculated with all samples. **(B)** Nannofossil species richness (Bown, 2005). **(C)** Coccolith mean size at the assemblage level during the Mesozoic (Aubry et al., 2005) and the Cenozoic (Herrmann and Thierstein, 2012). Mesozoic coccolith mean size at the assemblage level is derived from a compilation of species-sizes as published in the literature (original taxonomic descriptions), to which the authors have added their own measurements of published material. This record consists of measurements of length and width of 302 species, which is about one third of all the described Mesozoic coccolith species. The grey area illustrates the minimum and maximum size recorded. Cenozoic coccolith mean size at the assemblage level is derived from measurements of entire coccolith assemblages during the last 66 Myr from a number of globally distributed deep-sea cores using automated scanning electron microscopy and image analysis processing. **(D)** $pCO_2$ through time with Foster et al. (2017; in ppm) compilation represented by black filled circles, Witkowski et al. (2018; in µatm) represented by grey open squares, and Miocene $pCO_2$ decline from Mejía et al. (2017) in white open circle with the range of possible values in grey. The LOESS trend was calculated on Foster's and Witkowski's data assuming ppm and µatm are equivalent, and is represented by a black line and grey envelop representing the 95% confidence interval around the calculated trend.

Two time intervals are geographically well-documented (Fig. S3), mostly in European sites, the Toarcian (Early Jurassic; ~183-174 Ma) and the Valanginian (Early Cretaceous; ~140-133 Ma). NAR paleomaps have been constructed, based on averaged NAR-values for each site in both time intervals. During the Toarcian, NAR is higher in northern shallow epicontinental seas than in southeastern Tethyan open-sea (Fig. 3a). Conversely, during the Valanginian, NAR is higher in tropical open-seas than in northeastern European epicontinental seas near the Viking Corridor (i.e., the connection between Boreal and European seas; Westermann, 1993) (Fig. 3b-c). Finally, the highest Toarcian NAR (located in France and Yorkshire) is similar to the lowest Valanginian NAR (located in Greenland, North Sea and France) (Table S2). Compared to nannofossil species-richness and coccolith mean size, these results open new insights into the evolution of calcareous nannoplankton over the past ~200 Ma. Three distinct phases can be observed.

During the Jurassic and Early Cretaceous, the smoothed NAR increased alongside with species richness (Bown, 2005) while coccolith size was steadily small and started to increase at the Jurassic/Cretaceous boundary (Aubry et al., 2005). The beginning of this phase is marked by high calcareous nannoplankton production in epicontinental seas, whereas the end of this phase is marked by greater production in tropical open-ocean environments, as shown by the NAR maps (Fig. 3). Hence, this Invasion phase reflects a ~80 Myr-long gradual invasion of western Tethys and Atlantic Oceans by calcareous nannoplankton during the Jurassic-Early Cretaceous time interval. *Watznaueria barnesiae* (i.e. a cosmopolitain Mesozoic coccolithophore species) coccolith biometric data show only one size population between Western Tethys (La Charce-Vergol, France) and Western Panthalassa (ODP1149, Nadezhda basin close to Japan) during the Lower Cretaceous. Thus mixing of coccolithophore population was effective at that time testifying for a continuous genetic flux through the Tethys following a circum-global circulation (Gollain et al., 2019). The genetic flux was probably sustained for many calcareous nannoplankton between both region of the oceans at that time. The Early Cretaceous Invasion phase observed in the Atlantic Ocean may have thus happened in all open oceans realms worldwide, although our restricted European/North Hemisphere dataset cannot corroborate it.

An abrupt change in NAR dynamics, which is steadily-high since the Early Cretaceous (~120 Ma) according to the LOESS trend, marks the beginning of the second phase. From this point up to the end of the Cretaceous, NAR remained high but the nannofossil species-richness and the coccolith mean size increased since the beginning of the Cretaceous following the Cope-Depéret's rule (i.e. increase in coccolith size over evolutionary time; Aubry et al., 2005). As seen in the Valanginian NAR maps (Fig. 3B-C), by this time the shift in calcareous nannoplankton production toward the open-seas was already accomplished. This phase corresponds to the Specialization phase, where more and more species shared an increasingly filled ecospace through specialization to particular ecological niches."

After the Cretaceous-Paleogene (i.e., K–Pg) mass extinction event, calcareous nannoplankton recovered following the same two phases, namely Invasion and Specialization, but on a short time interval (less than 4 Myr) although our Paleocene NAR record is too limited to unambiguously confirm this pattern. Finally, a last phase in the calcareous nannoplankton evolution started in the Eocene (Fig. 2; Fig. S3) with smoothed NAR steadily high or slightly increasing (Fig. 2; Fig. S1) but the nannofossil species-richness and coccolith mean size both tending to decrease. This may correspond to an Establishment phase

where less species with smaller sizes predominated. This Establishment phase reached a climax in modern oceans with the dominance within the coccolithophore community of the iconic small-sized species *Emiliania huxleyi* (e.g. Ziveri et al., 2000; Baumann et al., 2004).

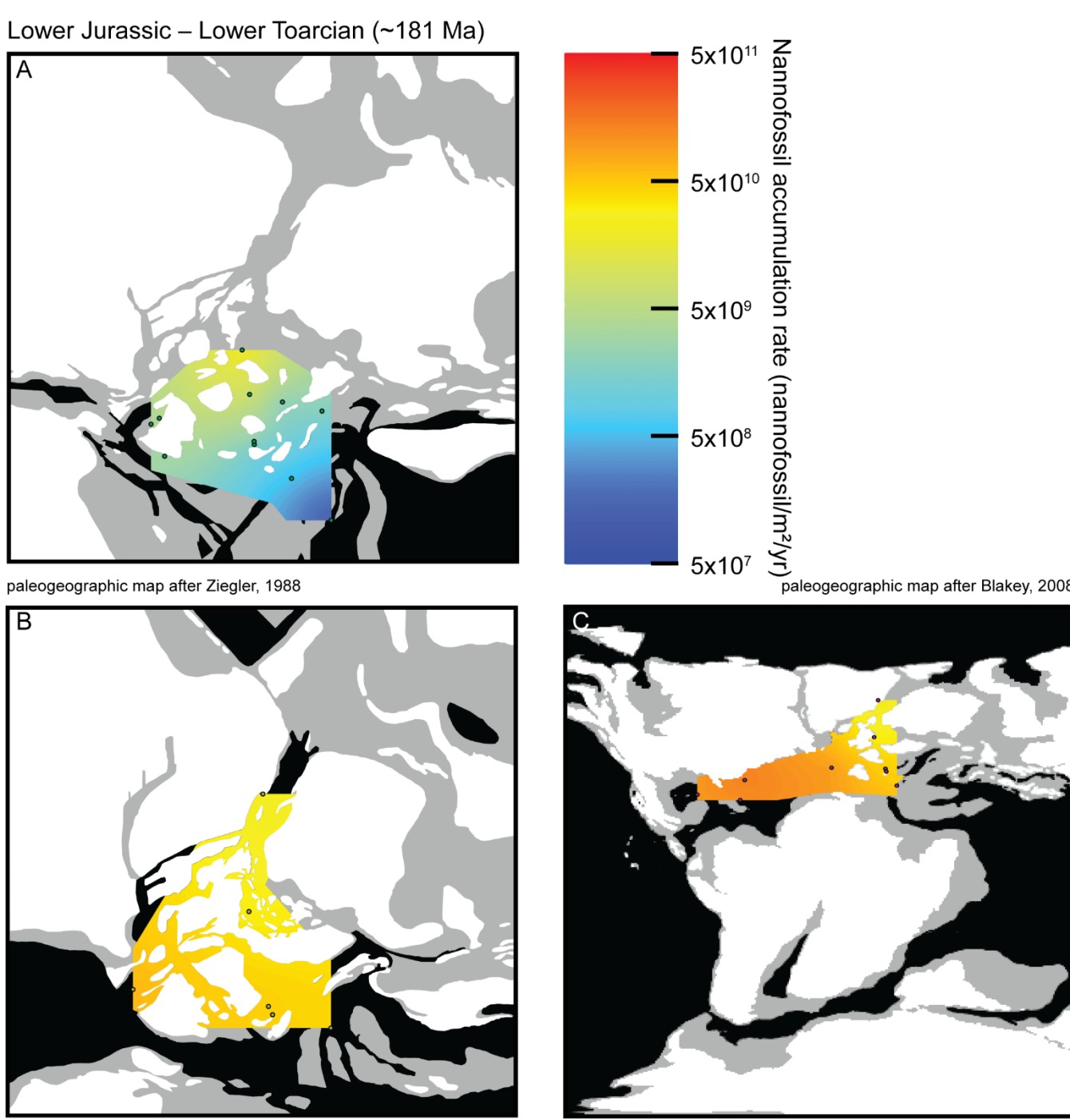

**Lower Jurassic – Lower Toarcian (~181 Ma)**

Nannofossil accumulation rate (nannofossil/m²/yr)

$5 \times 10^{11}$

$5 \times 10^{10}$

$5 \times 10^{9}$

$5 \times 10^{8}$

$5 \times 10^{7}$

paleogeographic map after Ziegler, 1988

paleogeographic map after Blakey, 2008

**Lower Cretaceous – Valanginian (~135 Ma)**

**Lower Cretaceous – Valanginian (~135 Ma)**

**Figure 3: Maps of nannofossil accumulation rate (nannofossil/m²/yr) drawn from the linear interpolation of the measurements realized in various sites (Table S2). Emerged lands are drawn in white; epicontinental seas are indicated in grey, and open oceans are indicated in black. (a) Toarcian in Europe, 11 sites considered, namely: Peniche (Portugal), Rabaçal (Portugal), La Cerradura (Spain), HTM-102 (France), Tournadous (France), Saint-Paul-des-Fonts (France), Yorkshire (UK), Dotternhausen (Germany),**

Somma (Italy), Chionistra (Greece), and Réka Valley (Hungary), for a total of 229 analyzed samples. (b) Valanginian in Europe, 6 sites considered, namely: Perisphinctes ravine (Greenland), ODP638 (North Atlantic), Vergol-La Charce (France), Carajuan (France), BGS 81/43 (North Sea), and Polaveno (Italy), for a total of 371 analyzed samples. (c) Valanginian in Europe and the Atlantic, Ocean adding 3 sites to the European ones: DSDP535 (Mexico Gulf), DSDP534A (North Atlantic), and DSDP603B (North Atlantic), for a total of 517 analysed samples. Paleogeographic maps modified from Ziegler (1988) and Blakey (2008).

## 4. Discussion

Extant calcareous nannoplankton is neither uniformly nor randomly dispersed in the global ocean (e.g. Winter et al., 1994). Its distribution in ecological niches is shaped by (i) abiotic parameters such as temperature, salinity, pH, and water mixing, but also by the availability in nutrients or light (e.g. Margalef, 1978; Balch, 2004), and (ii) by functional interactions with other organisms such as viruses (Frada et al., 2008), phytoplankton and grazers (Litchman et al., 2006). Extant coccolithophores are commonly viewed as "intermediate" organisms in *Margalef's mandala* (i.e. Fig. 2 from Margalef, 1978), so basically transitional between *K-* (corresponding to organisms evolving in more stable, predicable and saturated environments; e.g. dinoflagellates) and *r-* (organisms living in unstable, non-predictable, and unsaturated environments; e.g. diatoms) *strategists* (Reznick et al., 2002), living in intermediate nutrient-concentration waters, turbulence and light availability (Margalef, 1978; Balch, 2004; Tozzi et al., 2004). Thus, the integration of abiotic and biotic parameters explains the macroecological distribution of extant calcareous nannoplankton. Macroecological pattern may change through time due to evolution. And long-term evolution – macroevolution – is influenced by abiotic and biotic drivers that have given two evolutionary models: Red Queen (Van Valen, 1973) and Court Jester (Barnosky, 2001) hypotheses. The former states that biotic interactions drive evolutionary changes, whereas the latter asserts that changes in physical environments initiate evolutionary changes. The different phases observed here could be described in the light of macroecological and macroevolutionary models.

The Invasion phase during the Jurassic-Early Cretaceous is marked by both increasing NAR and nannoplankton species richness, indicating that the new occurring species increases the NAR without limiting the distribution of already existing species. Hence, the ecology of Jurassic-Early Cretaceous nannoplankton species was closer to the "*r-strategist*" pole of density-independent selection (Reznick et al., 2002). The Invasion phase echoes the beginning of the diversification of various planktic organisms – the *Mesozoic Plankton Revolution*. Even if there is an important change in the organization of the plankton community, the Invasion phase is likely driven by the Pangea breakup. This major tectonic event gave origin to newly formed oceanic domains, created perennial connections between the Pacific, Tethys and Atlantic oceans, and initiated sea-level rise and flooding of continental areas, finally establishing more numerous and heterogeneous ecological niches (Roth, 1989; Katz et al., 2004). The Mesozoic change in ocean chemistry, with increase in Cd, Cu, Mo, Zn and nitrate availability linked to deep-ocean oxygenation, would also have favored the development of the red lineage algae (i.e. using chlorophylle a, with chlorophylle c and fucoxanthin as accessory pigments typical in Haptophyte) such as coccolithophores (Falkowski et al., 2004). Although this Court Jester scenario most likely explains the invasion of the oceans by calcareous nannoplankton, Suchéras-Marx and co-workers pointed out that the increase in NAR during the Early Bajocian (Middle Jurassic; ~170 Ma) could also

have resulted from a more efficient exploitation of ecological niches by the newly originated species (Suchéras-Marx et al., 2015).

The following Specialization phase is marked by calcareous nannoplankton species having reached the maximum production of platelets (on average, ~$10^{11}$ nannofossils/m$^2$/yr), but these were produced by an increasing number of species characterized

by a higher coccolith size variance than in the previous phase (Fig. 2). This record suggests that more and more species shared an increasingly filled ecospace, therefore becoming more specialized to peculiar ecological niches. This specialization might correspond to an adaptation of different species to a particular ecological niche, to variable trophic levels (i.e., oligo- to eutrophic; e.g., Herrle, 2003; Lees et al., 2005), or temperature conditions (e.g. Mutterlose et al., 2014), or seasonality and blooming (e.g. Thomsen, 1989). Consequently, late Early and Late Cretaceous species were closer to the "*K-strategist*" pole

of density-dependent selection, corresponding to organisms evolving closer to carrying capacity. This time interval witnessed many drastic short-term climatic and environmental perturbations such as OAEs, thermal optimums or cooling (Friedrich et al., 2012), but also some relatively stable long-term physical conditions (e.g., sea-level; Müller et al., 2008). Despite some major short-term abiotic parameters changes, this macroevolutionary phase is more compatible with the Red Queen model. This time interval is the paroxysm of the *Mesozoic Plankton Revolution* with the first occurrence of diatoms, a plateau of marine

dinoflagellate species-richness, and the diversification of planktic foraminifera which, together with calcareous nannoplankton (Falkowski et al., 2004; Knoll and Follows, 2016), contributed to form massive chalk deposits (Roth, 1986). These various lines of evidence point toward an increase in interaction and competition between plankton organisms. Ultimately, the *Mesozoic Plankton Revolution* led to a bottom-up control of plankton on the entire marine ecosystem structure (Knoll and Follows, 2016), as revealed by the diversification of spatangoids echinoids, palaeocorystids crabs, Ancyloceratina ammonites

(Fraaije et al., 2018) and many other groups during the *Mesozoic Marine Revolution* (Vermeij, 1977) including highly diverse marine reptiles (Pyenson et al., 2014).

The Court Jester model well applies to the K–Pg mass extinction (66 Ma). This mass extinction event is related to abiotic perturbation, i.e. the Deccan traps volcanism (Courtillot et al., 1986), and an asteroid impact (Alvarez et al., 1980). This event had a catastrophic impact on calcareous nannoplankton diversity with a species turnover up to 80 % during the crisis (Bown,

2005). The K–Pg crisis almost shut down the pelagic production, raw NAR values returning to Lower Jurassic ones ($10^7$-$10^8$ nannofossils/m$^2$/yr; Fig. 2) (Hull et al., 2011). Our record of the aftermath of the K–Pg event indicates that the NAR recovered to pre-extinction levels in less than 4 Myr (Fig. 2). Associated with this Paleocene post-crisis NAR increase, which was followed by a steady production for the rest of this Epoch, an increase in coccolith mean size and in species-richness is observed (Fig. 2). At a much shorter time-scale, the Paleocene appears therefore similar to the Jurassic-Cretaceous interval in that a first

Invasion phase (the post-crisis biotic recovery) and the origination of new calcareous nannoplankton families (Bown, 2005) is followed by a period of species diversification and ecological specialization – a Specialization phase.

Eventually, at the end of the Paleogene and during the Neogene, calcareous nannoplankton experienced a third evolutionary phase – the Establishment – characterized by high raw NAR, and lower species richness involving smaller sized species than

in the Cretaceous. This last phase may have been driven by combined abiotic and biotic changes. First, the decrease in $pCO_2$ below a threshold throughout the Neogene could have driven the decrease in coccolithophore cell-size (Hannisdal et al., 2012) based on estimation of coccolith size decrease (Bolton et al., 2016). The carbon supply to coccolithophore cells is indeed sustained by $CO_2$ diffusion through the cellular membrane and depends on the cell surface/volume ratio, which is in turn

controlled by cell size. In many coccolithophores, there is a linear (isometric) relation between coccolith-size and cell-size (Henderiks, 2008). Consequently, the fitness decrease of large-sized species related to the $pCO_2$ drawdown led to a reduction in species richness. Secondly, diatoms tremendously diversified due to increase in silicic acid input to the oceans during this time interval (Spencer-Cervato, 1999; Cermeño et al., 2015), locally outcompeting calcareous nannoplankton, and only the most competitive coccolithophore species continued to proliferate. A habitat partitioning resulted, with calcareous

nannoplankton dominating the open-ocean oligotrophic areas, whereas diatoms thrived in meso-eutrophic coastal regions (Margalef, 1978). Nevertheless, modern-day calcareous nannoplankton is still more abundant in eutrophic upwelling regions than in open-oceans (Baumann et al., 2004), underscoring a complex rearrangement of microplankton community rather than a simple replacement of calcareous nannoplankton by diatoms.

## 5. Conclusion

Coccolithophores represent about half of the calcium carbonate in late Holocene deep-sea sediments but they were less abundant at the onset of calcareous nannoplankton evolution. Since the first occurrence of calcareous nannoplankton in the Late Triassic, the colonization of the oceans was a long-lasting and gradual process which can be separated in three successive phases, based on comparison of the nannofossil accumulation rate, species-richness and coccolith mean size variations. The

first phase from Early Jurassic to Early Cretaceous, corresponds to the nannoplankton oceans' Invasion. This phase is marked by an increasing NAR trend and in species richness along with steady to slight increase in coccolith mean size. In this time interval, our results suggest that the nannofossil accumulation almost exclusively occurred in epicontinental seas. By the Early Cretaceous, a phase of Specialization started. NAR attained the highest values while species-richness and coccolith mean size continued to increase. Moreover, NAR became highest in open-ocean tropical environments. During this second phase, an

increasing number of species tended to specialize and to share more efficiently the available ecospace. After the K–Pg mass extinction that led to a new and brief Invasion and Specialization phase, a third, and ongoing phase began during the Eocene-Oligocene. It is marked by a steady NAR but reduced species richness and coccolith mean size. A smaller number of species characterized by smaller size produce as many fossil coccoliths as before, pointing toward an increase in absolute abundances, at least for some species. This Establishment phase may be simultaneously related to the diversification and competitive

interaction of diatoms and to a decrease in atmospheric $pCO_2$. Finally, the long-term calcareous nannoplankton evolution over the past 200 Myr appears as a gradual colonization of almost all marine environments within the World Ocean. Such colonization was successively shaped by abiotic and biotic factors ultimately pointing toward the Court Jester and the Red

Queen macroevolutionary models as likely scenarii of the Invasion and Specialization phases, respectively, and both models apply to the more recent Establishment phase.

## 6. Data availability

Data are available in SI in two excel files. Table S1 gathered the dataset of nannofossil accumulation rate in the different settings studied in this work, sorted in chronological order. Each sheet presents the location of the site, the age (relative and absolute), the nannofossil absolute abundance, the sedimentation rate, the nannofossil accumulation rate, and other information such as the sample name, height in the section and the published reference. Table S2 gathered dataset of nannofossil accumulation rate used to construct Fig. 3. The table presents for both considered geological stages (i.e., Toarcian and Valanginian) the location of each site, their mean nannofossil absolute abundance and mean nannofossil accumulation rate, and the number of sample per site.

## 7. Sample availability

Slides made by BSM and EM for calcareous nannofossil study are curated at the Collections de Géologie de Lyon de l'Université Lyon 1 (collection code FSL).

## 8. Supplement link

Supplementary information linked to this manuscript are available. Supplementary text associated with supplementary figures S1 to S4 and supplementary references of compiled data.

## 9. Author contribution

BSM and EM contributed equally to this work. BSM and EM designed the study and compiled the data; BP provided accumulation rates; BSM did the calculations; PA constructed the maps; EM, FG and JP provided unpublished data; BSM, EM and FG devised the model, with advises from GE. BSM, EM, FG and GE wrote the text, with significant inputs from all other authors.

## 10. Competing interests

The authors declare that they have no conflict of interest.

## 11. Acknowledgements

EM acknowledges funding from INSU SYSTER 2011-2012 and INTERRVIE 2014-2015. BSM kindly thanks Jorijntje Henderiks for constructive comments on a previous version of the data set.

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
