# Peer review of "The colonization of the oceans by calcifying pelagic algae"

_Biogeosciences, 2018_

## Short Comment (SC1) · 15 Feb 2019

The authors might also like to consider the manuscript by Kvale et al., currently under review for the same journal (pending acceptance) https://www.biogeosciences-discuss.net/bg-2018-467/

---

## Referee Comment (RC1) · Anonymous Referee #1 · 21 Mar 2019

Dear Editor,

The manuscript by Suchéras-Marx et al. presents an interesting compilation of calcareous nannoplankton accumulation rate records through the Jurassic to Neogene. Combined with earlier published compilations of nannofossil species richness and coccolith mean size through this interval, these records provide interesting insights in the macroevolutionary patterns involved in the colonization of the world oceans by calcareous nannoplankton. The manuscript is generally well-written and concise, and presents an interesting discussion on the observed evolutionary dynamics.

Perhaps the only major point of concern is the geographical limits of the data set. For the Jurassic and Cretaceous (representing a very long time interval, which happens to be crucial for the colonization of the world oceans by calcareous nannoplankton),

basically all of the data sets are from the Northern Atlantic & Western Europe. While I realize that the authors are limited by the availability of data sets, I think this is a major weakness of the presented compilation, and a point that is not sufficiently addressed in the manuscript. The authors should include a bit of discussion on the possible problems and/or complications with their data set. How sure are the authors that the Northern Atlantic/Western European records are representative for the global oceans? The authors talk about "open oceans" of the Valanginian, but the Northern Atlantic is still relatively isolated by these times. We know absolutely nothing about the real open oceans (the Pacific & the eastern Tethys). Is it possible that the recorded patterns are diachronous between different ocean basins? I think this merits at least a little bit of discussion, also if the authors do believe they can build a case based on the Northern Atlantic and Western European records alone.

Likewise, here and there, the authors oversimplify things a little bit too much, in my opinion. For example, describing the Early Cretaceous to Late Cretaceous, a period of ∼60 million years(!), with major pulses of mid ocean spreading, oceanic anoxic events, soaring atmospheric pCO2 concentrations, major climate shifts, major evolutionary developments in many biological groups, as "characterized by a relatively stable environmental setting" is perhaps a bit to simplistic. It feels a bit like a reversed argumentation: "because we see evolutionary patterns that are compatible with the Red Queen macroevolutionary model, the changes in the physical environments must be limited." While I follow the authors in that the influences of the biotic interactions (forcing a Red Queen type evolution) were probably stronger in this time interval compared to the influences of abiotic environmental changes, it is way to simplistic to state that, therefore, the 60 million year period of the Early Cretaceous to Late Cretaceous was characterized by stable environmental conditions. . . The authors could, and should, rephrase these kind of statements, to incorporate a bit more nuances.

One more thing: I highly recommend using more than one pCO2 reconstruction for this time interval (the Jurassic to Neogene). The authors have chosen to use the compilation presented in Hönisch et al. (2012) as there sole atmospheric pCO2 record, while this particular reconstruction seems to underestimate the pCO2 concentrations for the mid-Cretaceous (a crucial interval for the present study). These kind of problems can be circumvented by using more than one compilation or model-based reconstruction, averaging out potential problems with any particular reconstruction.

To conclude, with a little bit more nuancing and a bit more discussion on the possible problems and pitfalls, this manuscript has the potential to be an important contribution to the field. Therefore, I believe manuscript merits a publication in Biogeosciences, after some major revisions.

Attached to this review is a list with comments and suggestions.

Please also note the supplement to this comment:
https://www.biogeosciences-discuss.net/bg-2018-493/bg-2018-493-RC1-supplement.pdf
* * *
[Figure]

**Supplement:**

**Page 4 Lines 9-15:** Is it possible that the recorded patters are diachronous between different ocean basins? How sure are the authors that the Northern Atlantic/Western European records are representative for the global oceans? I think this merits at least a little bit of discussion.

**Figure 2:**

It is unfortunate that such a large portion of the recorded patterns are forced by the Northern Atlantic/Western European records. Perhaps the authors could color-code the datapoints in A to show the regions where these datapoints are derived from?

In addition, I am a bit troubled by the atmospheric CO2 reconstruction used in this study (based on the compilation in Hönisch et al. 2012). It is odd that the lower Jurassic values are so much higher than the mid-Cretaceous values, while we know that the mid-Cretaceous (Cenomanian-Turonian) was characterized by exceptionally high pCO2 concentrations. I believe the authors should have a look at some other pCO2 reconstructions, for example the one recently published by Witkowski et al (2018) in Science Advances; or the modelling work by e.g. Dana Royer. Including more compilations and reconstructions would greatly improve the manuscript.

**Page 7:** Is it possible that these patterns are only representative for the depicted region (the North Atlantic & Western Europe)? Can the authors argue why they believe the patterns in this rather limited (and restricted) region are representative for the global oceans?

**Page 7 Line 3:** I am not familiar with the term "Viking Corridor". Can the authors explain this? Or provide a reference to a study that does?

**Page 7 line 7**: in the record of Aubry et al. (2005), the coccolith size actually already starts increasing in the Late Jurassic. In addition, I wonder, why would Cope-Depéret's rule not yet be in place in the Middle Jurassic?

**Page 7 Lines 10-11 "*Hence, this Invasion phase reflects a ~80 Myr-long gradual invasion of world open oceans by calcareous nannoplankton during the Jurassic-Early Cretaceous time interval.*" =>** It is interesting to see how the radiation/invasion over this interval directly and indirectly, led to a proliferation of various benthic groups such as burying and swimming crabs and irregular echinoids as well as nektonic groups such as ancyloceratine heteromorph ammonites. See the study of Fraaije et al (2018) for this. Perhaps worth mentioning?

**Page 7 Lines 16-17**: can the authors elaborate a little bit more on which type of specializations they are talking here? Which kind of ecological conditions?

**Page 7 lines 22-23:** How does this work? What forces this "establishement phase"? I see that the authors discuss this topic further on in the manuscript, but in its current form, this sentence triggers the big "why?" question. Why did less species, with smaller sizes, dominate? What forces this>?

**Page 9 Line 28 "*within less than..*"**: this "within" feels a bit superfluous. Maybe just "in less than.."?

**Page 9 Line 32**: Why is "Specialization" capitalized here?

**Page 10 Lines 15-22 "This phase was not related to major physical or chemical changes, climatic and environmental parameters showing steady-state dynamics":** With major pulses of mid ocean spreading, oceanic anoxic events, soaring pCO2 concentrations, major climate shifts, major evolutionary devolopments in many biological groups, the Early Cretaceous to Late Cretaceous, a period of ~60 million years (!), can hardly be called "*a relatively stable environmental setting*". I suggest the authors rephrase this paragraph.

**Page 10 Lines 22-23:** => this bottom-up control of the marine ecosystem structuration also led to the emergence and dispersion in the different higher-tier trophic levels, discussed earlier (Fraaije et al. 2018).

**Page 11 Lines 3-4:** Perhaps the authors can elaborate a little on why the diatoms diversified over this time interval? This group appears to have shown an adaptive radiation tied to higher dissolved silica concentrations and stronger circulation and upwelling from the mid-Cenozoic onwards (Falkowski et al., 2004).

**Page 11, Lines 15-16** *"The first phase, Early Jurassic to Early Cretaceous, corresponds to the nannoplankton oceans' Invasion marked by an increase in NAR and in species richness along with a quite steady coccolith mean size."* This sentence is difficult to follow. Please rewrite.

---

## Referee Comment (RC2) · 16 Apr 2019

The manuscript "The colonization of the oceans by calcifying pelagic algae" by B. Sucheras-Marx et al. describes colonization of the oceans by coccolithophorids since the last 200 M. This well written manuscript is based on the compilation of nannoplankton accumulation rates in sediments, brought in context with previously published species richness, coccolith size as well as atmospheric CO2. Results indicate a colonization of the oceans in distinct phases, shaped by the reproduction strategy, interactions with other planktonic organisms and the physical environment. The research is original and provides interesting findings to the community. The data set compilation seems to have been carried out with great care, even though, sadly, the available data is confined largely to the Atlantic, therefore I would suggest to maybe

rephrase the main conclusions of the manuscript from "World Oceans" to "Atlantic". The manuscript is concisely written, however, could benefit from a re-organization of the Discussion paragraph in my opinion, so that each phase is discussed in its own paragraph, instead of discussing the colonization twice in 4.1 and 4.2. I have some reservations regarding the smoothing of the NAR and the seemingly arbitrary reference to sometimes the smoothed trend and sometimes the underlying raw data. The authors should carefully re-examine each time the NAR is discussed and elaborate on when which datatype is discussed (see major comments in the attached review). I would recommend publication of this manuscript after minor revisions have been carried out. I wish the authors good luck with the revisions and remain available for further feedback and discussions.

Please also note the supplement to this comment:
https://www.biogeosciences-discuss.net/bg-2018-493/bg-2018-493-RC2-supplement.pdf

**Supplement:**

The manuscript "The colonization of the oceans by calcifying pelagic algae" by B. Sucheras-Marx et al. describes colonization of the oceans by coccolithophorids since the last 200 M. This well written manuscript is based on the compilation of nannoplankton accumulation rates in sediments, brought in context with previously published species richness, coccolith size as well as atmospheric CO2. Results indicate a colonization of the oceans in distinct phases, shaped by the reproduction strategy, interactions with other planktonic organisms and the physical environment.

The research is original and provides interesting findings to the community. The data set compilation seems to have been carried out with great care, even though, sadly, the available data is confined largely to the Atlantic, therefore I would suggest to maybe rephrase the main conclusions of the manuscript from "World Oceans" to "Atlantic". The manuscript is concisely written, however, could benefit from a re-organization of the Discussion paragraph in my opinion, so that each phase is discussed in its own paragraph, instead of discussing the colonization twice in 4.1 and 4.2.

I have some reservations regarding the smoothing of the NAR and the seemingly arbitrary reference to sometimes the smoothed trend and sometimes the underlying raw data. The authors should carefully re-examine each time the NAR is discussed and elaborate on when which datatype is discussed (see major comments below).

I would recommend publication of this manuscript after minor revisions have been carried out. I wish the authors good luck with the revisions and remain available for further feedback and discussions.

Please see my comments below (p=page, l=line):

Major comments:

NAR calculation: Since the majority of the manuscript hinges on the NAR, it would be great if the authors could provide an propagation of error for the NAR values, as they are calculated from 3 other variables. Additionally the NAR in Fig. 2 has a high variability of several orders of magnitude, can the authors elaborate on this a bit, e.g. is this caused by pooling different ocean locations, where changes could have happened at a different point in time?

Smoothed curve versus raw data: Currently, in some time periods smoothed NAR values are discussed and sometimes the raw data. Please state each time, which data is taken (raw data or smoothed trend). Please be careful in not mixing the two.

e.g. p9 l29 " a steady production for the rest of the epoch" seem to be rather subjective, as there seems to be rather a huge variability in observed NAR post K-Pg until the end of the Paleocene, just the chosen smoothing factor results in a steady NAR.  How have the authors assessed "stable phases" in NAR versus "changing phases" of NAR? Only by visual observation of the smoothed trend?

By just looking at the smoothed curve, variability in the NAR data is lost. While I agree that in some time points a SF of 0.1 is influenced by the sampling resolution, however, in other

time points variability and trends are lost by a higher smoothing factor (e.g. the increase in NAR since the middle Paleogene, which is "smoothed away" otherwise).

Furthermore (p9 l27) here the average NAR shows no change during the K.Pg event, but NAR clearly changes, which is also discussed.

Layout Figure 2: please mark the individual colonization phases in a way, that they are easy to be put into context with the NAR record. Currently, the phases are indicated on the far right and the NAR record is on the far left, making it hard to see the exact phase changes. I would suggest shading of the background. Please also indicate the Torcian and Valangian. And add a line for the K-Pg event, as some of the statements (e.g. p9 l28 " ..the NAR recovered to pre-extinction levels within less than 4 Myr") are hard to follow with the current Figure layout.

Minor Comments:

p2 l2: represent (without s)

p2 l6-13: also refer to the Kuenen Event in the discussion or remove from Introduction

p3 l17 $mm^2$

p4 Fig. 1 caption: type of outcrop: rephrase outcrop; deep sea drilling is not an outcrop

p5 l5: SI= suppl. inform. (define).

p7 l 6: I would structure the paragraph according to the different phases, e.g. add a break in the middle of l. 6.

p7 l 14: regarding the versatile readership of BGD, I would refrain from using too many specific terms such as *Cope-Deperets* rule, which are not explained in the Introduction, same for *Margalefs mandala* in p9 l12, also explain briefly K and r strategists (for readers from a more geological background).

p7 l 17: ecospace or ecological niche?

p7 l 24: dominance: rephrase, as modern oceans are not dominated by *Ehux*, but it is the dominant cocco

p8 Fig 3: please add also a time stamp to panel c (Valanginian?)

p9: when the term *species* is used, calc. nannoplankton species is meant? or

coccolithophorids?

p 9: I find the terms *R-pole* and *K-pole* confusing, are these commonly used terms? Or do they just hint towards the respective areas in Margalefs mandala?

p9 l21: the maximum occurs much later, this need to be rephrased

p9 l24: please explain "roughly stable"

p9 l. 32: where is the "ecological specialization" seen in the data?

p10 l10: What are "red lineage algae"? Those belonging to the Red Queen Model?

p10 l 18 - 20: please add citations

p 10 l 28: where is the "post crisis Invasion period" in Fig 2?

p10 l 31: "smaller sized species than in the Mesozoic": to me it looks like the average coccolith size is relatively the same between this period and the Jurassic portion of the Mesozoicum

p11 l 1 The "decrease in pCO2" during the Neogene is not visible in Fig2, maybe another dataset would be more suitable? Also, how do the authors then explain the stable coccolith mean size and increasing NAR during the Jurassic, where pCO2 showed the largest drop?

---

## Author Comment (AC1) · 27 May 2019

The comments are organized as :

Comment from R1

→ Our answer with pasted modification in the text and the relevant page and lines modified (or figure).

Dear Editor, The manuscript by Suchéras-Marx et al. presents an interesting compilation of calcareous nannoplankton accumulation rate records through the Jurassic to Neogene. Combined with earlier published compilations of nannofossil species richness and coccolith mean size through this interval, these records provide interesting insights in the macroevolutionary patterns involved in the colonization of the world

oceans by calcareous nannoplankton. The manuscript is generally well-written and concise, and presents an interesting discussion on the observed evolutionary dynamics.

→ We thank Reviewer 1 for his positive remarks and his constructive review. We have now addressed the main concerns and suggestions. Please find below the detailed responses.

Perhaps the only major point of concern is the geographical limits of the data set. For the Jurassic and Cretaceous (representing a very long time interval, which happens to be crucial for the colonization of the world oceans by calcareous nannoplankton), basically all of the data sets are from the Northern Atlantic & Western Europe. While I realize that the authors are limited by the availability of data sets, I think this is a major weakness of the presented compilation, and a point that is not sufficiently addressed in the manuscript. The authors should include a bit of discussion on the possible problems and/or complications with their data set. How sure are the authors that the Northern Atlantic/Western European records are representative for the global oceans? The authors talk about "open oceans" of the Valanginian, but the Northern Atlantic is still relatively isolated by these times. We know absolutely nothing about the real open oceans (the Pacific & the eastern Tethys). Is it possible that the recorded patterns are diachronous between different ocean basins? I think this merits at least a little bit of discussion, also if the authors do believe they can build a case based on the Northern Atlantic and Western European records alone.

→ We have pondered this point. We highlight this point in section 2.3 and in section 3 (p.8).

Likewise, here and there, the authors oversimplify things a little bit too much, in my opinion. For example, describing the Early Cretaceous to Late Cretaceous, a period of ~60 million years(!), with major pulses of mid ocean spreading, oceanic anoxic events, soaring atmospheric $pCO_2$ concentrations, major climate shifts, major evolutionary developments in many biological groups, as "characterized by a relatively stable environmental setting" is perhaps a bit to simplistic. It feels a bit like a reversed argumentation: "because we see evolutionary patterns that are compatible with the Red Queen macroevolutionary model, the changes in the physical environments must be limited." While I follow the authors in that the influences of the biotic interactions (forcing a Red Queen type evolution) were probably stronger in this time interval compared to the influences of abiotic environmental changes, it is way to simplistic to state that, therefore, the 60 million year period of the Early Cretaceous to Late Cretaceous was characterized by stable environmental conditions: The authors could, and should, rephrase these kind of statements, to incorporate a bit more nuances.

→ We have added a bit of nuance in the section 4 (p.11).

One more thing: I highly recommend using more than one pCO2 reconstruction for this time interval (the Jurassic to Neogene). The authors have chosen to use the compilation presented in Hönisch et al. (2012) as there sole atmospheric pCO2 record, while this particular reconstruction seems to underestimate the pCO2 concentrations for the mid-Cretaceous (a crucial interval for the present study). These kind of problems can be circumvented by using more than one compilation or model based reconstruction, averaging out potential problems with any particular reconstruction.

→ We changed Hönisch et al. 2012 for Foster et al., 2017 and Witkowski et al., 2018 and Mejia et al., 2017 (for the Late Miocene) in Fig.2.

To conclude, with a little bit more nuancing and a bit more discussion on the possible problems and pitfalls, this manuscript has the potential to be an important contribution to the field. Therefore, I believe manuscript merits a publication in Biogeosciences, after some major revisions.

Page 4 Lines 9-15: Is it possible that the recorded patters are diachronous between different ocean basins? How sure are the authors that the Northern Atlantic/Western European records are representative for the global oceans? I think this merits at least

a little bit of discussion.

→ This part was actually discussed in SI. Nevertheless, we have now highlighted this point in section 2.3 P4 L19-25 "The vast majority of the samples are from the Northern Hemisphere, and almost all samples for Jurassic and Cretaceous times are from Western Europe outcrops – a relatively poor quantitative record of nannofossils exists outside Europe and in oceanic sites issued from deep-sea drilling programs. Europe, North-Sea, Greenland and North Atlantic represent 81.16% of all compiled samples. Thus, results based on NAR in the Mesozoic will be mostly based on European/Atlantic localities and thus may describe pattern that occurred mainly in the Western Tethys and North Atlantic (see SI S3 and Fig. S3). For the Cenozoic, the data are more widely distributed but the sample per Myr is less abundant than in the Mesozoic (see SI S3 and Fig. S3)."

Figure 2: It is unfortunate that such a large portion of the recorded patterns are forced by the Northern Atlantic/Western European records. Perhaps the authors could color-code the datapoints in A to show the regions where these datapoints are derived from? In addition, I am a bit troubled by the atmospheric CO2 reconstruction used in this study (based on the compilation in Hönisch et al. 2012). It is odd that the lower Jurassic values are so much higher than the mid-Cretaceous values, while we know that the mid-Cretaceous (Cenomanian-Turonian) was characterized by exceptionally high pCO2 concentrations. I believe the authors should have a look at some other pCO2 reconstructions, for example the one recently published by Witkowski et al (2018) in Science Advances; or the modelling work by e.g. Dana Royer. Including more compilations and reconstructions would greatly improve the manuscript.

→ The pCO2 reconstruction from Hönisch et al., 2012 has been changed in Fig. 2 to Foster et al., 2017 and Witkowski et al., 2018, more recently updated. I have also added Mejia et al., 2017 for the Miocene (following Reviewer 2's comment). I would rather prefer not to add modelling works, there are too many solutions deriving from GEOCARB and updated, choosing one would be like choosing the one fitting

the best. In the new figure showing the NAR, now the "World" is grey and the Europe/NorthAtlantic is in black.

Page 7: Is it possible that these patterns are only representative for the depicted region (the North Atlantic & Western Europe)? Can the authors argue why they believe the patterns in this rather limited (and restricted) region are representative for the global oceans?

→ We cannot demonstrate that the pattern observed is worldwide, the variability in NAR in North Atlantic and western Tethys is so large for a given age, that it might be different in other basins. We changed the text accordingly P8 L14-23 "Hence, this Invasion phase reflects a ∼80 Myr-long gradual invasion of western Tethys and Atlantic Oceans by calcareous nannoplankton during the Jurassic-Early Cretaceous time interval. According to coccolith biometric data, the Watznaueria barnesiae (i.e. a cosmopolitain Mesozoic coccolithophore species) genetic flux was maintained between populations in the western Tethys and in the eastern Panthalassa in the Lower Cretaceous likely related to circum-global circulation (Gollain et al., 2019) through the Tethys. The Early Cretaceous Invasion phase observed in the Atlantic Ocean may have thus happened in all open oceans realms worldwide, although our restricted European/North Hemisphere dataset cannot corroborate it."

Page 7 Line 3: I am not familiar with the term "Viking Corridor". Can the authors explain this? Or provide a reference to a study that does?

→ The Viking Corridor (Wertermann, 1993) or Viking Strait (Callomon, 1985) is the name of the connection between Boreal Sea and Northern Western Tethys and it is commonly used (e.g. Aberhan, 2001; Dera et al., 2009; Korte et al., 2015; Ruebsam et al., 2014). We have now added Westerman, 1993 reference in the text.

Page 7 line 7: in the record of Aubry et al. (2005), the coccolith size actually already starts increasing in the Late Jurassic. In addition, I wonder, why would Cope-Depéret's rule not yet be in place in the Middle Jurassic?

→ A small increase in the size trend starts around the J/K boundary. Hence, a net increase in size is observed since the base of the Early Cretaceous. The reason why the Cope-Deperet's rule does not apply to earlier times (e.g., the middle Jurassic) stays thus unclear. We have now added precisions in the text P8 L25-27 "From this point up to the end of the Cretaceous, NAR remained high but the nannofossil species-richness and the coccolith mean size increased since the beginning of the Cretaceous following the Cope-Depéret's rule (i.e. increase in size over evolutionary time; Aubry et al., 2005)."

Page 7 Lines 10-11 "Hence, this Invasion phase reflects a ~80 Myr-long gradual invasion of world open oceans by calcareous nannoplankton during the Jurassic-Early Cretaceous time interval." => It is interesting to see how the radiation/invasion over this interval directly and indirectly, led to a proliferation of various benthic groups such as burying and swimming crabs and irregular echinoids as well as nektonic groups such as ancyloceratine heteromorph ammonites. See the study of Fraaije et al (2018) for this. Perhaps worth mentioning?

→ We don't mention it in the results but we have now added it in fused Discussion (i.e. fusion of former sections 4.1 and 4.2) P12 L18-21 " Ultimately, the Mesozoic Plankton Revolution led to a bottom-up control of plankton on the entire marine ecosystem structure (Knoll and Follows, 2016), as revealed by the diversification of spatangoids echinoids, palaeocorystids crabs, Ancyloceratina ammonites (Fraaije et al., 2018) and many other groups during the Mesozoic Marine Revolution (Vermeij, 1977) including highly diverse marine reptiles (Pyenson et al., 2014)."

Page 7 Lines 16-17: can the authors elaborate a little bit more on which type of specializations they are talking here? Which kind of ecological conditions?

→ We have now changed these lines accordingly into P8 L29-30 "This phase corresponds to a Specialization phase, where more and more species shared an increasingly filled ecospace through specialization to particular ecological niches." This point

is more developed in the Discussion later in the text P12 L6-9 "This specialization might correspond to an adaptation of different species to a particular ecological niche, to variable trophic levels (i.e., oligo- to eutrophic; e.g., Herrle, 2003; Lees et al., 2005), or temperature conditions (e.g. Mutterlose et al., 2014), or seasonality and blooming (e.g. Thomsen, 1989)."

Page 7 lines 22-23: How does this work? What forces this "establishement phase"? I see that the authors discuss this topic further on in the manuscript, but in its current form, this sentence triggers the big "why?" question. Why did less species, with smaller sizes, dominate? What forces this?

→ We have now considered the possible reasons for this pattern and added discussion in the Discussion P13 L1-15. Following R2, the discussion has been thoroughly revised, by merging both sections 4.1 and 4.2. In the new version of the MS, the establishment phase is discussed in the last paragraph.

Page 9 Line 28 "within less than..": this "within" feels a bit superfluous. Maybe just "in less than.."?

→ corrected accordingly.

Page 9 Line 32: Why is "Specialization" capitalized here?

→ it shouldn't, corrected.

Page 10 Lines 15-22 "This phase was not related to major physical or chemical changes, climatic and environmental parameters showing steady-state dynamics": With major pulses of mid ocean spreading, oceanic anoxic events, soaring pCO2 concentrations, major climate shifts, major evolutionary developments in many biological groups, the Early Cretaceous to Late Cretaceous, a period of ∼60 million years (!), can hardly be called "a relatively stable environmental setting". I suggest the authors rephrase this paragraph.

→ corrected P12 L10-13 "This time interval witnessed many short-term climatic and

environmental perturbations such as OAEs, thermal optimums or cooling (Friedrich et al., 2012), but also some relatively stable long-term physical conditions (e.g., sea-level; Müller et al., 2008)."

Page 10 Lines 22-23: => this bottom-up control of the marine ecosystem structuration also led to the emergence and dispersion in the different higher-tier trophic levels, discussed earlier (Fraaije et al. 2018).

→ We have now added P12 L18-21 " Ultimately, the Mesozoic Plankton Revolution led to a bottom-up control of plankton on the entire marine ecosystem structure (Knoll and Follows, 2016), as revealed by the diversification of spatangoids echinoids, palaeocorystids crabs, Ancyloceratina ammonites (Fraaije et al., 2018) and many other groups during the Mesozoic Marine Revolution (Vermeij, 1977) including highly diverse marine reptiles (Pyenson et al., 2014)."

Page 11 Lines 3-4: Perhaps the authors can elaborate a little on why the diatoms diversified over this time interval? This group appears to have shown an adaptive radiation tied to higher dissolved silica concentrations and stronger circulation and upwelling from the mid-Cenozoic onwards (Falkowski et al., 2004).

→ We follow the latest argumentation about silicic acid input to the oceans (Cermeño, Falkowski et al., 2015 PNAS) P13 L9-10 "Secondly, diatoms tremendously diversified due to increase in silicic acid input to the oceans during this time interval (Spencer-Cervato, 1999; Cermeño et al., 2015) [. . .]"

Page 11, Lines 15-16 "The first phase, Early Jurassic to Early Cretaceous, corresponds to the nannoplankton oceans' Invasion marked by an increase in NAR and in species richness along with a quite steady coccolith mean size." This sentence is difficult to follow. Please rewrite.

→ We changed for P13 L22-24 "The first phase from Early Jurassic to Early Cretaceous, corresponds to the nannoplankton oceans' Invasion. This phase is marked by

an increase in NAR and in species richness along with steady to slight increase in coccolith mean size."

---

## Author Comment (AC2) · 27 May 2019

Answer to Nina Keul, reviewer 2

The comments are organized as :

Comment from R2

→ Anwer to R2

The manuscript "The colonization of the oceans by calcifying pelagic algae" by B. Sucheras-Marx et al. describes colonization of the oceans by coccolithophorids since the last 200 M. This well written manuscript is based on the compilation of nanno-plankton accumulation rates in sediments, brought in context with previously published

species richness, coccolith size as well as atmospheric CO2. Results indicate a colonization of the oceans in distinct phases, shaped by the reproduction strategy, interactions with other planktonic organisms and the physical environment. The research is original and provides interesting findings to the community. The data set compilation seems to have been carried out with great care, even though, sadly, the available data is confined largely to the Atlantic, therefore I would suggest to maybe rephrase the main conclusions of the manuscript from "World Oceans" to "Atlantic". The manuscript is concisely written, however, could benefit from a re-organization of the Discussion paragraph in my opinion, so that each phase is discussed in its own paragraph, instead of discussing the colonization twice in 4.1 and 4.2.

→ We thank Reviewer 2 for her positive comments. Following the R2's remarks, we have now merged sections 4.1 and 4.2 and re-organized the Discussion by paragraphs, the first one introducing the models, andthe following ones describing each phase in a separate paragraph.

I have some reservations regarding the smoothing of the NAR and the seemingly arbitrary reference to sometimes the smoothed trend and sometimes the underlying raw data. The authors should carefully re-examine each time the NAR is discussed and elaborate on when which datatype is discussed (see major comments below).

I would recommend publication of this manuscript after minor revisions have been carried out. I wish the authors good luck with the revisions and remain available for further feedback and discussions.

Please see my comments below (p=page, l=line): Major comments:

NAR calculation: Since the majority of the manuscript hinges on the NAR, it would be great if the authors could provide an propagation of error for the NAR values, as they are calculated from 3 other variables. Additionally the NAR in Fig. 2 has a high variability of several orders of magnitude, can the authors elaborate on this a bit, e.g. is this caused by pooling different ocean locations, where changes could have happened

at a different point in time?

→ We strongly agree with R2's remarks, I am myself fighting for more control on uncertainties in Earth Sciences in general (see as an example Suchéras-Marx et al., in revision at Marine Micropaleontology titled "Statistical confidence intervals for relative abundances and abundance-based ratios: simple practical solutions for an old overlooked question"). Unfortunately, in the present case, we cannot propagate uncertainties simply because the original publications do not provide information on the counting uncertainties, stratigraphic uncertainties or density. Thus, propagation would be completely arbitrary and may suggest to the reader that we control uncertainties, which is unfortunately not true.

Smoothed curve versus raw data: Currently, in some time periods smoothed NAR values are discussed and sometimes the raw data. Please state each time, which data is taken (raw data or smoothed trend). Please be careful in not mixing the two. e.g. p9 l29 " a steady production for the rest of the epoch" seem to be rather subjective, as there seems to be rather a huge variability in observed NAR post K-Pg until the end of the Paleocene, just the chosen smoothing factor results in a steady NAR. How have the authors assessed "stable phases" in NAR versus "changing phases" of NAR? Only by visual observation of the smoothed trend?

→ Yes, we do not calculate any test for that, we prefer to directly observe the curve and compare it to the other ones.

By just looking at the smoothed curve, variability in the NAR data is lost. While I agree that in some time points a SF of 0.1 is influenced by the sampling resolution, however, in other time points variability and trends are lost by a higher smoothing factor (e.g. the increase in NAR since the middle Paleogene, which is "smoothed away" otherwise). Furthermore (p9 l27) here the average NAR shows no change during the K/Pg event, but NAR clearly changes, which is also discussed.

→ R2 is right. In the new version of our MS, we now state when the raw NAR or the

smooth NAR is considered.

Layout Figure 2: please mark the individual colonization phases in a way, that they are easy to be put into context with the NAR record. Currently, the phases are indicated on the far right and the NAR record is on the far left, making it hard to see the exact phase changes. I would suggest shading of the background. Please also indicate the Torcian and Valangian. And add a line for the K-Pg event, as some of the statements (e.g. p9 l28 " ..the NAR recovered to pre-extinction levels within less than 4 Myr") are hard to follow with the current Figure layout.

→ Fig. 2 has been largely amended following R1 and R2 comments.

Minor Comments:

p2 l2: represent (without s)

→ corrected

p2 l6-13: also refer to the Kuenen Event in the discussion or remove from Introduction

→ the Kuenen Event is the shift from carbonate system dominated by neritic production from benthic organisms to pelagic production from planktic organisms. Hence it is crucial to cite this event in the Introduction because it is linked to planktic organisms' evolution. But because we don't quantify carbonate in this study, we cannot speculate on the timing of this event thus we don't discuss it. We have made some modification in the text to derive the point from Kuenen to evolution transition P1 L7-16 "There is then a transition from Jurassic calcareous nannofossil-poor to Late Cretaceous and Cenozoic calcareous nannofossil-rich oceanic sediments which has shifted the carbonate accumulation in neritic environment by benthic organisms to accumulation in pelagic environments by planktic organisms. This major carbonate system change is known as the Kuenen Event (Roth, 1989), and has been referred to a tectonically-mediated intensification of the ocean circulation. This event is concomitant with the development of several planktic groups (e.g., planktic foraminifera (Hart et al., 2003), diatoms (Kooistra

et al., 2007)), may be seen as a Mesozoic Plankton Revolution (derived from Vermeij, 1977) and thus is also dramatically related to plankton evolution. The causes and consequences of this biotic revolution have been extensively discussed, but the transition itself remains poorly documented; most interpretations solely rely on species richness (Falkowski et al., 2004; Knoll and Follows, 2016), which does not provide an exhaustive framework to fully appreciate the evolutionary history of calcareous nannoplankton."

p3 l17 mm2

→ corrected.

p4 Fig. 1 caption: type of outcrop: rephrase outcrop; deep sea drilling is not an outcrop

→ corrected.

p5 l5: SI= suppl. inform. (define)

→ corrected.

p7 l 6: I would structure the paragraph according to the different phases, e.g. add a break in the middle of l. 6.

→ corrected.

p7 l 14: regarding the versatile readership of BGD, I would refrain from using too many specific terms such as Cope-Deperets rule, which are not explained in the Introduction, same for Margalefs mandala in p9 l12, also explain briefly K and r strategists (for readers from a more geological background).

→ corrected; we have also added short precision of each specific terms listed above P8 L27 "[. . .] Cope-Depéret's rule (i.e. increase in size over evolutionary time; Aubry et al., 2005)"; P11 L11-12 "(i.e. Fig. 2 from Margalef, 1978)"; P11 L12-15 "between K- (corresponding to organisms evolving in more stable, predicable and saturated environments) and r- (organisms living in unstable, non-predictable, and unsaturated environments) strategists (Reznick et al., 2002), living in intermediate nutrient-concentration

waters, turbulence and light availability (Margalef, 1978; Balch, 2004; Tozzi et al., 2004)."

p7 l 17: ecospace or ecological niche?

→ ecospace.

p7 l 24: dominance: rephrase, as modern oceans are not dominated by Ehux, but it is the dominant cocco?

→ rephrased P9 L2-4 "This Establishment phase reached a climax in modern oceans with the dominance within the coccolithophore community of the iconic small-sized species Emiliania huxleyi (e.g. Ziveri et al., 2000; Baumann et al., 2004)."

p8 Fig 3: please add also a time stamp to panel c (Valanginian?)

→ slight modification of the Fig. 3 with a stratigraphic column.

p9: when the term species is used, calc. nannoplankton species is meant? Or coccolithophorids?

→ calcareous nannoplankton, the term is added to species when needed.

p 9: I find the terms R-pole and K-pole confusing, are these commonly used terms? Or do they just hint towards the respective areas in Margalefs mandala?

→ r- and K- strategists is a common term in ecology. We slightly modified it to highlight the fact that within plankton, calcareous nannoplankton are intermediate strategist and within calcareous nannoplankton there are species that are closer to one or the other side or "pole". We have now modified these lines to P11 L24-25 "Hence, the ecology of Jurassic-Early Cretaceous nannoplankton species was closer to the "r-strategist" pole of density-independent selection (Reznick et al., 2002). [. . .] P12 L9-10 Consequently, late Early and Late Cretaceous species were closer to the "K-strategist" pole of density-dependent selection, corresponding to organisms evolving closer to carrying capacity."

p9 l21: the maximum occurs much later, this need to be rephrased.

→ The sentence p9 l21 is: "It suggests that more and more species shared an increasingly filled ecospace (Fig. 2), therefore becoming more specialized to peculiar environmental conditions." I am sorry but I don't understand what R2 means.

p9 l24: please explain "roughly stable"

→ bad writing. Modified for "The raw NAR reached an optimum at ∼133 Ma and the smooth NAR is flat from ∼117 Ma until the K–Pg mass extinction event (66 Ma), which had a catastrophic impact on calcareous nannoplankton diversity with a species turnover up to 80 % during the crisis (Bown, 2005)."

p9 l. 32: where is the "ecological specialization" seen in the data?

→ the specialization is not seen in data per say but inferred based on the observation that more species sharing the same ecospace are producing a stable number of individuals. If the ecospace is split between more species, then the species must be more specialized to peculiar conditions.

p10 l10: What are "red lineage algae"? Those belonging to the Red Queen Model?

→ the red lineage algae are those using chlorophyll c and derivatives as accessory pigments. Added in text P11 L31-33 "[. . .] red lineage algae (i.e. using chlorophylle a, with chlorophylle c and fucoxanthin as accessory pigments typical in Haptophyte) such as coccolithophores (Falkowski et al., 2004)"

p10 l 18 - 20: please add citations.

→ citations added P12 L14-17 "This time interval is the paroxysm of the Mesozoic Plankton Revolution with the first occurrence of diatoms, a plateau of marine dinoflagellate species-richness, and the diversification of planktic foraminifera which, together with calcareous nannoplankton (Falkowski et al., 2004; Knoll and Follows, 2016), contributed to form massive chalk deposits (Roth, 1986)."

p 10 l 28: where is the "post crisis Invasion period" in Fig 2?

→ we can't see it, we just speculate it must be one since the diversity and nannoplankton productivity dramatically dropped at the K-Pg boundary. We have added precision in the following P12 L29-32 "At a much shorter time-scale, the Paleocene appears therefore similar to the Jurassic-Cretaceous interval in that a first Invasion phase (the post-crisis biotic recovery) and the origination of new calcareous nannoplankton families (Bown, 2005) is followed by a period of species diversification and ecological specialization – a Specialization phase."

p10 l 31: "smaller sized species than in the Mesozoic": to me it looks like the average coccolith size is relatively the same between this period and the Jurassic portion of the Mesozoicum.

→ modified to "Cretaceous".

p11 l 1 The "decrease in pCO2" during the Neogene is not visible in Fig2, maybe another dataset would be more suitable? Also, how do the authors then explain the stable coccolith mean size and increasing NAR during the Jurassic, where pCO2 showed the largest drop?

→ Firstly, the Bolton's model is based on reaching an unknown threshold in pCO2 during the Miocene with pCO2 too low to sustain CO2 diffusion through the cell wall for both organic and inorganic carbon fixation in large coccolithophore cell. The Lower Jurassic decrease just not reach this threshold. We have added P13 L4 "below a threshold" to the sentence. Secondly, the Miocene pCO2 discussion is really critical and controversial between specialists. In order to overcome this issue we have changed Hönisch's data compilation in Fig.2 to Foster's data compilation (which is really similar but more recent), added Witkowski et al. 2018 data (asked by R1) and added in grey Mejia et al. 2017 results (range due to uncertainties) which record the Miocene pCO2 decrease.

---

## Author Comment (AC4) · 27 May 2019

**This PDF file includes:**
Supplementary Text
Figures and captions S1-S4
Captions for Tables S1-S2
Supplementary references

**S1. Data set compilation**

All data compiled are provided in an Excel file, with one sheet per site or paper (Table S1) for a total of 3895 data points across 79 sites or papers. The compiled sites are:

- Sao Pedro de Moel, Portugal, Late Sinemurian–Lower Pliensbachian (Plancq et al., 2016)

- Peniche, Portugal, Early Pliensbachian (Mattioli, unpublished data)

- Peniche, Portugal, Late Pliensbachian (Reggiani et al., 2010)

- La Cerradura, Spain, Late Pliensbachian–Early Toarcian (Reolid et al., 2014)

- Peniche, Portugal, Early Toarcian (Mattioli et al., 2009; Suan et al., 2008)

- Tournadous, France, Late Pliensbachian–Early Toarcian (Mailliot et al., 2009; Suan et al., 2008)

- Saint-Paul-des-Fonts, France, Late Pliensbachian–Early Toarcian (Mailliot et al., 2009; Suan et al., 2008)

- Somma, Italy, Late Pliensbachian–Early Toarcian (Mattioli et al., 2009)

- Dotternhausen, Germany, Early Toarcian (Mattioli et al., 2009)

- Yorkshire, UK, Early Toarcian (Plancq, 2009)

- Réka Valley, Hungary, Early Toarcian (Plancq, 2009)

- Chionistra, Greece, Early Toarcian (Kafousia et al., 2014)

- HTM-102, France, Early Toarcian (Mattioli et al., 2009)

- K2-5, France, Early Toarcian (Plancq, 2009)

- Rabaçal, Portugal, Late Pliensbachian–Late Toarcian (Kenjo, 2010)

- Cabo Mondego, Portugal, Late Aalenian–Early Bajocian (Suchéras-Marx et al., 2013; Suchéras-Marx et al., 2012)

- Chaudon-Norante, France, Late Aalenian–Early Bajocian (Suchéras-Marx et al., 2013; Suchéras-Marx et al., 2014)

- La Voulte, France, Middle Callovian–Early Oxfordian (Giraud, unpublished data)

- La Voulte, France, Middle Oxfordian (Excoffier, 2001; Pittet, 2006)

- Meussia, France, Middle Oxfordian (Excoffier, 2001; Pittet, 2006)

- Balingen–Tieringen, Germany, Late Oxfordian (Mattioli, unpublished data; Pittet, 2006)

- Plettenberg, Germany, Late Oxfordian (Olivier et al., 2004; Pittet, 2006)

- Le Pas de l'Assassin, France, Late Oxfordian–Early Kimmeridgian (Carcel, 2009; Carcel et al., 2010)

- DSDP105, North Atlantic, Tithonian–Valanginian (Bornemann et al., 2003)

- DSDP534A, North Atlantic, Tithonian–Valanginian (Bornemann et al., 2003)

- DSDP367, North Atlantic, Tithonian–Valanginian (Bornemann et al., 2003; Lancelot and Seibold, 1978)

- Perisphinctes Ravine, Greenland, Late Ryazanian–Late Hauterivian (Pauly et al., 2012)

- Rødryggen, Greenland, Late Ryazanian–Late Hauterivian (Pauly et al., 2012)

- Polaveno, Italy, Late Berriasian–Early Hauterivian (Erba and Tremolada, 2004; Gréselle and Pittet, 2010)

- DSDP534A, North Atlantic, Late Berriasian–Early Hauterivian (Bornemann et al., 2005; Gréselle and Pittet, 2010)

- DSDP603B, North Atlantic, Late Berriasian–Late Valanginian (Bornemann et al., 2005; Gréselle and Pittet, 2010)

- Vergol–La Charce, France, Valanginian (Gréselle and Pittet, 2010; Gréselle et al., 2011)

- Carajuan, France, Early Valanginian (Gréselle and Pittet, 2010; Riquier, 2002)

- DSDP535, Mexico Gulf, Early Valanginian–Early Hauterivian (Kessels et al., 2006)

- ODP638, North Atlantic, Early Valanginian–Early Hauterivian (Kessels et al., 2006)

- BGS81/43, North Atlantic, Early Valanginian–Early Hauterivian (Kessels et al., 2006)

- BGS81/43, North Sea, Late Valanginian–Late Hauterivian (Williams and Bralower, 1995)

- Speeton, UK, Early Hauterivian–Barremian (Williams and Bralower, 1995)

- Otto Gott, Germany, Barremian (Mutterlose, 1998; Williams and Bralower, 1995)

- Nora-1, Danemark, Barremian (Mutterlose and Bottini, 2013)

- North Jens-1, Danemark, Late Barremian–Early Aptian (Mutterlose and Bottini, 2013)

- A39-Braunschweig, Germany, Early Barremian–Early Aptian (Pauly et al., 2013)

- Takal Kuh, Iran, Early Aptian (Mahanipour et al., 2011)

- Notre-Dame-de-Rosans, France, Aptian (Giraud et al., 2018)

- Pré-Guittard, France, Late Aptian (Herrle, 2002; Herrle et al., 2003)

- Alma, Morocco, Late Aptian–Early Albian (Peybernes et al., 2013)

- Addar, Morocco, Late Aptian–Early Albian (Peybernes et al., 2013)

- Tamzergout, Morocco, Late Aptian–Early Albian (Peybernes et al., 2013)

- Hyèges, France, Late Aptian (Giraud, unpublished data; Herrle, 2002)

- L'Arboudeysse, France, Early Albian (Herrle, 2002)

- DSDP545, North Atlantic, Early Albian (Herrle, 2002)

- Col de Palluel, France, Late Albian (Bornemann et al., 2005)

- Blieux, France, Early Cenomanian–Middle Cenomanian (Giraud et al., 2013; Reboulet et al., 2013)

- Wunstorf, Germany, Middle Cenomanian–Middle Turonian (Linnert et al., 2010; Voigt et al., 2008)

- ODP1258, Central Atlantic, Early Cenomanian–Middle Turonian (Hardas and Mutterlose, 2007; Shipboard Scientific Party, 1985)

- ODP1260, Central Atlantic, Early Cenomanian–Early Turonian (Hardas and Mutterlose, 2007; Shipboard Scientific Party, 1985)

- Holywell Pinnacles, UK, Late Cenomanian–Early Turonian (Linnert et al., 2011b; Voigt et al., 2008)

- DSDP549-551, North Atlantic, Middle Cenomanian–Late Maastrichtian (Linnert et al., 2011a; Shipboard Scientific Party, 1985)

- Kronsmoor, Germany, Late Campanian–Early Maastrichtian (Linnert et al., 2016)

- Qreiya 1, Egypt, Danian–Selandian (Youssef Ali, 2009)

- Qreiya 3, Egypt, Danian–Selandian (Sprong et al., 2011)

- Araas, Egypt, Danian (Youssef Ali, 2009)

- Duwi, Egypt, Danian (Youssef Ali, 2009)

- ODP1260B, Central Atlantic, Late Paleocene–Early Eocene (Mutterlose et al., 2007; Youssef Ali and Mutterlose, 2004)

- Wadi Abu Ghurra, Egypt, Late Maastrichtian–Early Eocene (Youssef Ali and Mutterlose, 2004)

- Kurkur Naqb Dungul, Egypt, Late Paleocene–Early Eocene (Youssef Ali and Mutterlose, 2004)

- ODP1263, South Atlantic, Late Paleocene–Early Eocene (Zuzlewski, 2014)

- ODP1209A, North Pacific, Eocene (Salaviale, 2013)

- DSDP511, South Atlantic, Rupelian–Priabonian (Plancq et al., 2014)

- DSDP516, South Atlantic, Oligocene–Miocene (Plancq et al., 2012; Plancq et al., 2013)

- DSDP608, North Atlantic, Oligocene–Miocene (Plancq et al., 2013)

- DSDP588C, North Atlantic, Oligocene–Miocene (Plancq et al., 2013)

- ODP752A, Indian Ocean, Miocene–Pleistocene (Suchéras-Marx and Henderiks, 2012)

- DSDP525, South Atlantic, Miocene–Pleistocene (Suchéras-Marx and Henderiks, 2012)

- ODP806B, East Pacific, Miocene–Pleistocene (Suchéras-Marx and Henderiks, 2012)

- ODP707A, Indian Ocean, Miocene–Pleistocene (Suchéras-Marx and Henderiks, 2012)

- ODP982B, North Atlantic, Miocene–Pleistocene (Suchéras-Marx and Henderiks, 2012)

- Punta di Maiata, Italy, Zanclean (Mattioli, unpublished data)

- Punta Grande/Punta Piccola, Italy, Pliocene (Plancq et al., 2015)

**S2. Impact of LOESS smoothing factor on the long-term trend of nannofossil accumulation rate.** Interpretation of the long-term trend can be arguably linked to the smoothing factor (SF) selected for the LOESS (LOcally wEighted Scatterplot Smoothing). In Fig. S1 we present the nannofossil accumulation rate for SF = 0.1, 0.25, 0.5, 0.75, and 1. The smaller the smoothing factor, the rougher and less marked the smoothing is. Whereas the smoothed curve at SF 0.1 is clearly influenced by the sampling resolution along the analyzed time series, from SF 0.25 to SF 1 the same long-term trend is observed without short-term artifact. Accordingly, the discussed trend appears to be a reliable long-term pattern unrelated to the selected SF-value.

[Figure]

**Fig. S1.** Comparison of the LOESS smoothing for nannofossil accumulation rates (nannofossil/m²/yr) with five different smoothing factor values (from left to right): SF 0.1, SF 0.25, SF 0.5, SF 0.75, and SF 1. SF 0.5 is the one discussed in the main text.

[Figure]

**Fig. S2.** Comparison of microplankton evolution and environmental/climatic changes through time. Dinoflagellate cyst richness (number of species in black and genera in grey) (Kooistra et al., 2007; Stover et al., 1996); diatoms richness (number of species in black and genera in grey) (Spencer-Cervato, 1999), nannoplankton species richness (Bown, 2005); planktic foraminifera species richness (Peters et al., 2013); coccolith mean size at the assemblage level (μm, Mesozoic (Aubry et al., 2005) and Cenozoic (Herrmann and Thierstein, 2012)); nannofossil accumulation rate (this study); atmospheric $CO_2$ (ppm) (Foster et al., 2017; Mejía et al., 2017); compilation of OAEs, thermal events and K/Pg mass extinction; Phanerozoic sea-level (m) (Hardenbol et al., 1998); $\delta^{13}C$ (‰, VPDB; square: belemnite; open circle: planktic foraminifera; black circle: benthic foraminifera) (Friedrich et al., 2005; Prokoph et al., 2008); $\delta^{18}O$ (‰, VPDB; square: belemnite; open circle: planktic foraminifera; black circle: benthic foraminifera) (Friedrich et al., 2005; Prokoph et al., 2008); phosphorus accumulation rate (mg/m²/yr) (Föllmi, 1995); Mg/Ca (molar ratio), Ca (mEq/L), calcite vs. aragonite seas (Stanley, 2008). The dinoflagellate species richness presents the same broad pattern as calcareous nannofossil species richness with an increase up to the Early Cretaceous and a decrease from the end of the Paleocene up to Holocene (Falkowski et al., 2004). This long-term pattern is interrupted by a two-phase decrease: one during the early Late Cretaceous and another during the K/Pg crisis. The long-term variations of diatoms species and genera richness are completely different from those observed for both calcareous nannofossils and dinoflagellates cysts. Unfortunately, the lack of literature data on diatoms and dinoflagellate accumulation rates hampers further comparisons with the evolutionary pattern discussed for calcareous nannoplankton. The record of $\delta^{13}C$ and phosphorus accumulation rates is not discussed in the main text. They do not show long-term changes, but rather shorter cyclic variations intimately related to climate changes which are tuned by orbital cycles. On the long-term scale, there is no relationship between calcareous nannoplankton evolution and $\delta^{13}C$ or phosphorus accumulation rates, but there are likely relationships on the short-term scale (e.g. Mattioli et al., 2009) which are not captured in the present record.

**S3. Disparity of sampling through time and space.** The database presented in this study gathered 3895 samples from 79 sites over the last 193 Myr. This large amount of data is not evenly distributed through time. Figure S3 presents the number of sample per Myr and the number of sites per Myr. Usually, the more sites compiled, the more samples there are in the database, although this is not observed for the Neogene. This figure also highlights the discrete sampling pattern, with some time intervals being more densely studied than others. Three time-intervals are highly documented: the early Toarcian (Early Jurassic), the Valanginian (Early Cretaceous), and the Cenomanian/Turonian transition (Late Cretaceous). To a lesser extent, we can also cite the Aptian-Albian transition (Early Cretaceous) and the Paleogene.

The maps in Fig. S3 present the sites studied for the four geological Periods considered (Quaternary excluded). The Jurassic samples are mainly represented by European sites and are all located in the Northern Hemisphere. The Cretaceous samples are equably distributed between European and Atlantic deep-sea drilling sites, all of them being from the Northern Hemisphere. The Paleogene samples are from Egyptian sites and four deep-sea drilling sites, three from the Central-South Atlantic and one from the North Pacific. Finally, the Neogene samples are dominated by deep-sea drilling sites from the Southern Hemisphere. The analyzed database is not uniform through time, with a shift from European and Northern Hemisphere Mesozoic sites to Southern Hemisphere samples in the Neogene.

[Figure]

**Fig. S3.** Number of samples and sites per Myr and sites location for the four Periods studied.

**S4. Impact of sedimentary rates on the computation of nannofossil accumulation rate.** This study discusses nannofossil accumulation rate which is derived from nannofossil absolute abundance and sedimentary rates (see Methods). All absolute abundance data compiled but one paper (Erba and Tremolada, 2004) have been obtained using the same method; methodological biases are thus negligible at this level. Conversely, the sedimentary rates used to calculate nannofossil accumulation rates derive from the International Chronostratigraphic Chart 2012 (Gradstein et al., 2012) or from cyclostratigraphy-based estimated durations. These two different methods used to calculate nannofossil accumulation rates might impact the short-term fluctuations observed, but they are not likely to affect the long-term trends discussed here. Figure S4 compares the long-term trends (LOESS SF 0.5) for nannofossil absolute abundance (nannofossil/$g_{bulk}$) and nannofossil accumulation rate (nannofossil/m²/yr). The comparison of these two plots highlights some differences due to local sedimentary patterns influencing the nannofossil accumulation rate (e.g., during the Early Cretaceous); nevertheless, the overall long-term trend discussed in the article is the same between nannofossil absolute abundance and accumulation rate. This observation allows us to rule out a bias effect of sedimentary rate calculation on the resulting long-term trend.

[Figure]

**Fig. S4.** Comparison between the compiled nannofossil absolute abundances (nannofossil/$g_{bulk}$) and nannofossil accumulation rates (nannofossil/m²/yr) computed in this study.

**Table S1.** Dataset of nannofossil accumulation rate in the different settings studied in this work, sorted in chronological order. Each sheet presents the location of the site, the age (relative and absolute), the nannofossil absolute abundance, the sedimentation rate, the nannofossil accumulation rate, and other information such as the sample name, height in the section and the published reference.

**Table S2.** Dataset of nannofossil accumulation rate used to construct Fig. 3. The table presents for both considered geological stages (i.e., Toarcian and Valanginian) the location of each site, their mean nannofossil absolute abundance and mean nannofossil accumulation rate, and the number of sample per site.

**S5. Supplementary references for nannofossil absolute abundance, nannofossil accumulation rate and sedimentation rate calculation compiled in this study, and supplementary references in Fig. S2 (by alphabetical order)**

Bornemann, A., Aschwer, U., and Mutterlose, J.: The impact of calcareous nannofossils on the pelagic carbonate accumulation across the Jurassic-Cretaceous boundary, Palaeogeogr. Palaeoclimatol. Palaeoecol., 199, 187-228, 2003.

Bornemann, A., and Mutterlose, J.: Calcareous nannofossil and d13C records from the early Cretaceous of the western Atlantic ocean: evidence for enhanced fertilization across the Berriasian-Valanginian transition, Palaios, 23, 821-832, 2008.

Bornemann, A., Pross, J., Reichelt, K., Herrle, J. O., Hemleben, C., and Mutterlose, J.: Reconstruction of short-term palaeoceanographic changes during the formation of the Late Albian 'Niveau Breistroffer' black shales (Oceanic Anoxic Event 1d, SE France), J. Geol. Soc. London, 162, 623-639, http://dx.doi.org/10.1144/0016-764903-171, 2005.

Carcel, D.: Caractérisation des environnements de dépôt dominés par les tempêtes, PhD, Université Claude Bernard Lyon1, 131 pp., 2009.

Carcel, D., Colombié, C., Giraud, F., and Courtinat, B.: Tectonic and eustatic control on a mixed siliciclastic-carbonate platform during the Late Oxfordian-Kimmeridgian (La Rochelle platform, western France), Sediment. Geol., 223, 334-359, 2010.

Excoffier, F. : Modèle d'interaction climat–niveau marin et le signal carbonate dans les sédiments hémipélagiques du Mésozoïque (Oxfordien, Valanginien, SE France), Master Thesis, Université Claude Bernard Lyon1, 42 pp., 2001.

Föllmi, K. B.: 160 m.y. record of marine sedimentary phosphorus burial: Coupling of climate and continental weathering under greenhouse and icehouse conditions, Geology, 23, 859-862, 1995.

Friedrich, O., Herrle, J. O., and Hemleben, C.: Climatic changes in the Late Campanian-Early Maastrichian: Micropaleontological and stable isotopic evidence from an epicontinental sea, J. Foraminifer. Res., 35, 228-247, http://dx.doi.org/10.2113/35.3.228, 2005.

Friedrich, O., Norris, R. D., and Erbacher, J.: Evolution of middle to Late Cretaceous oceans-A 55 m.y. record of Earth's temperature and carbon cycle, Geology, 40, 107-110, 2012.

Giraud, F.: Unpublished data – La Voulte, France.

Giraud, F.: Unpublished data – Hyèges, France.

Giraud, F., Pittet, B., Grosheny, D., Baudin, F., Lécuyer, C., and Sakamoto, T.: The palaeoceanographic crisis of the Early Aptian (OAE 1a) in the Vocontian Basin (SE France), Palaeogeogr. Palaeoclimatol. Palaeoecol., https://doi.org/10.1016/j.palaeo.2018.09.014, 2018.

Giraud, F., Reboulet, S., Deconinck, J.-F., Martinez, M., Carpentier, A., and Bréziat, C.: The Mid-Cenomanian Event in southeastern France: Evidence from palaeontological and clay mineralogical data, Cretac. Res., 46, 43-58, 2013.

Gréselle, B., and Pittet, B.: Sea-level reconstructions from the Peri-Vocontian Zone (Southeast France) point to Valanginian glacio-eustasy, Sedimentology, 57, 1640-1684, 2010.

Gréselle, B., Pittet, B., Mattioli, E., Joachimski, M., Barbarin, N., Riquier, L., Reboulet, S., and Pucéat, E.: The Valanginian isotope event: A complex suite of palaeoenvironmental perturbations, Palaeogeogr. Palaeoclimatol. Palaeoecol., 306, 41-57, 2011.

Hardas, P., and Mutterlose, J.: Calcareous nannofossil assemblages of Oceanic Anoxic Event 2 in the equatorial Atlantic: Evidence of an eutrophication event, Mar. Micropaleontol., 66, 52-69, 2007.

Hardenbol, J., Thierry, J., Farley, M. B., Jacquin, T., De Graciansky, P.-C., and BVail, P. R.: Mesozoic and Cenozoic sequence stratigraphy of European basins, SEPM Spec. Publ., 60, 1-13, 1998.

Herrle, J. O.: Paleoceanographic and Paleoclimatic Implications on Mid-Cretaceous Black Shale Formation in the Vocontian Basin and the Atlantic: Evidence from Calcareous Nannofossils and Stable Isotopes, Tubinger Mikropaläontologische Mitteilungen, 27, 144, 2002.

Herrle, J. O., Pross, J., Friedrich, O., and Hemleben, C.: Short-term environmental changes in the Cretaceous Tethyan Ocean: micropalaeontological evidence from the Early Albian Oceanic Anoxic Event 1b, Terra Nova, 15, 14-19, 2003.

Kafousia, N., Karakitsios, V., Mattioli, E., Kenjo, S., and Jenkyns, H. C.: The Toarcian Oceanic Anoxic Event in the Ionian Zone, Greece, Palaeogeogr. Palaeoclimatol. Palaeoecol., 393, 135-145, http://dx.doi.org/10.1016/j.palaeo.2013.11.013, 2014.

Kenjo, S.: Biostratigraphie à nannofossiles calcaires et changements paléoenvironnemantaux au Toarcien. L'exemple du Bassin Lusitanien (Portugal), Master Thesis, Université Claude Bernard Lyon1, 50 pp, 2010.

Kessels, K., Mutterlose, J., and Michalzik, D.: Early Cretaceous (Valanginian - Hauterivian) calcareous nannofossils and isotopes of the northern hemisphere: proxies for the understanding of Cretaceous climate, Lethaia, 39, 157-172, http://dx.doi.org/10.1080/00241160600763925, 2006.

Lancelot, Y., and Seibold, E.: The Evolution of the Central Northeastern Atlantic—Summary of Results of DSDP Leg 41, DSDP Reports and Publications XLI, 1215-1245, 1978.

Linnert, C., Engelke, J., Wilmsen, M., and Mutterlose, J.: The impact of the Maastrichtian cooling on the marine nutrient regime—Evidence from midlatitudinal calcareous nannofossils, Paleoceanography, 31, 2015PA002916, http://dx.doi.org/10.1002/2015PA002916, 2016.

Linnert, C., Mutterlose, J., and Erbacher, J.: Calcareous nannofossils of the Cenomanian/Turonian boundary interval from the Boreal Realm (Wunstorf, northwest Germany), Mar. Micropaleontol., 74, 38-58, 2010.

Linnert, C., Mutterlose, J., and Herrle, J. O.: Late Cretaceous (Cenomanian-Maastrichtian) calcareous nannofossils from Goban Spur (DSDP Sites 549, 551): Implications for the palaeoceanography of the proto North Atlantic, Palaeogeogr. Palaeoclimatol. Palaeoecol., 299, 507-528, 2011a.

Linnert, C., Mutterlose, J., and Mortimore, R.: Calcareous nannofossils from Eastbourne (Souteastern England) and the paleoceanography of the Cenomanian-Turonian boundary interval, Palaios, 26, 298-313, 10.2110/palo.2010.p10-130r, 2011b.

Mahanipour, A., Mutterlose, J., Kani, A. L., and Adabi, M. H.: Palaeoecology and biostratigraphy of early Cretaceous (Aptian) calcareous nannofossils and the $\delta^{13}C_{carb}$ isotope record from NE Iran, Cretac. Res., 32, 331-356, http://dx.doi.org/10.1016/j.cretres.2011.01.006, 2011.

Mailliot, S., Mattioli, E., Bartolini, A., Baudin, F., Pittet, B., and Guex, J.: Late Pliensbachian-Early Toarcian (Early Jurassic) environmental changes in an epicontinental basin of NW Europe (Causses area, central France): A micropaleontological and geochemical approach, Palaeogeogr. Palaeoclimatol. Palaeoecol., 273, 346-364, 2009.

Mattioli, E.: Unpublished data – Balingen–Tieringen, Germany.

Mattioli, E.: Unpublished data – Peniche, Portugal.

Mattioli, E.: Unpublished data – Punta de Maiata, Italy.

Mattioli, E., Pittet, B., Petitpierre, L., and Mailliot, S.: Dramatic decrease of pelagic carbonate production by nannoplankton across the Early Toarcian anoxic event (T-OAE), Glob. Planet. Change, 65, 134-145, 2009.

Mutterlose, J.: The Lower and Upper Cretaceous of the Hannover-Braun-schweig area (NW Germany), in: Key localities of the Northwest European Cretaceous, edited by: Mutterlose, J., Bornemann, A., Rauer, S., Spaeth, C., and Wood, C. J., Bochumer geologische und geotechnische Arbeiten, Bochum, 81-90, 1998.

Mutterlose, J., and Bottini, C.: Early Cretaceous chalks from the North Sea giving evidence for global change, Nat. Commun., 4, 1686, http://dx.doi.org/10.1038/ncomms2698, 2013.

Mutterlose, J., Linnert, C., and Norris, R.: Calcareous nannofossils from the Paleocene-Eocene Thermal Maximum of the equatorial Atlantic (ODP Site 1260B): Evidence for tropical warming, Mar. Micropaleontol., 65, 13-31, 2007.

Olivier, N., Pittet, B., and Mattioli, E.: Palaeoenvironmental control on sponge-microbialite reefs and contemporaneous deep-shelf marl-limestone deposition (Late Oxfordian, southern Germany), Palaeogeogr. Palaeoclimatol. Palaeoecol., 212, 233-263, 2004.

Pauly, S., Mutterlose, J., and Alsen, P.: Early Cretaceous palaeoceanography of the Greenland-Norwegian Seaway evidenced by calcareous nannofossils, Mar. Micropaleontol., 90-91, 72-85, 2012.

Pauly, S., Mutterlose, J., and Wray, D. S.: Palaeoceanography of Lower Cretaceous (Barremian-Lower Aptian) black shales from northwest Germany evidenced by calcareous nannofossils and geochemistry, Cretac. Res., 42, 28-43, http://dx.doi.org/10.1016/j.cretres.2013.01.001, 2013.

Peters, S. E., Kelly, D. C., and Fraass, A. J.: Oceanographic controls on the diversity and extinction of planktonic foraminifera, Nature, 493, 398-403, 2013.

Peybernes, C., Giraud, F., Jaillard, E., Robert, E., Masrour, M., Aoutem, M., and Içame, N.: Stratigraphic framework and calcareous nannofossil productivity of the Essaouira-Agadir

Basin (Morocco) during the Aptian-Early Albian: Comparison with the north-Tethyan margin, Cretac. Res., 39, 149-169, 2013.

Pittet, B.: Les alternances marno–calcaires ou l'enregistrement de la dynamique de production et d'export des plates–formes carbonatées, HDR Thesis, Université Claude Bernard Lyon1, 79 pp, 2006.

Plancq, J.: Caractéristiques de la phase de récupération par le nannoplancton calcaire après l'événement anoxique du Toarcien inférieur (Jurassique inférieur), Master Thesis, Université Claude Bernard Lyon1, 50 pp., 2009.

Plancq, J., Grossi, V., Henderiks, J., Simon, L., and Mattioli, E.: Alkenone producers during late Oligocene-early Miocene revisited, Paleoceanography, 27, PA1202, 2012.

Plancq, J., Grossi, V., Pittet, B., Huguet, C., Rosell-Melé, A., and Mattioli, E.: Multi-proxy constraints on sapropel formation during the late Pliocene of central Mediterranean (southwest Sicily), Earth Planet. Sci. Lett., 420, 30-44, http://dx.doi.org/10.1016/j.epsl.2015.03.031, 2015.

Plancq, J., Mattioli, E., Henderiks, J., and Grossi, V.: Global shifts in Noelaerhabdaceae assemblages during the late Oligocene-early Miocene, Mar. Micropaleontol., 103, 40-50, http://dx.doi.org/10.1016/j.marmicro.2013.07.004, 2013.

Plancq, J., Mattioli, E., Pittet, B., Baudin, F., Duarte, L. V., Boussaha, M., and Grossi, V.: A calcareous nannofossil and organic geochemical study of marine palaeoenvironmental changes across the Sinemurian/Pliensbachian (early Jurassic, ~ 191 Ma) in Portugal, Palaeogeogr. Palaeoclimatol. Palaeoecol., 449, 1-12, http://dx.doi.org/10.1016/j.palaeo.2016.02.009, 2016.

Plancq, J., Mattioli, E., Pittet, B., Simon, L., and Grossi, V.: Productivity and sea-surface temperature changes recorded during the late Eocene-early Oligocene at DSDP Site 511 (South Atlantic), Palaeogeogr. Palaeoclimatol. Palaeoecol., 407, 34-44, http://dx.doi.org/10.1016/j.palaeo.2014.04.016, 2014.

Prokoph, A., Shields, G. A., and Veizer, J.: Compilation and time-series analysis of a marine carbonate d18O, d13C, 87Sr/86Sr and d34S database through Earth history, Earth Sci. Rev., 87, 113-133, http://dx.doi.org/10.1016/j.earscirev.2007.12.003, 2008.

Riquier, L.: Variations spatio–temporelles des assemblages de nannoplancton calcaire sur un transect palte–forme externe/bassin épicontinental dans le bassin Vocontien (Crétacé inférieur; SE France), Master Thesis, Université Claude Bernard Lyon1, 51 pp, 2002

Reboulet, S., Giraud, F., Colombié, C., and Carpentier, A.: Integrated stratigraphy of the Lower and Middle Cenomanian in a Tethyan section (Blieux, southeast France) and correlations with Boreal basins, Cretac. Res., 40, 170-189, 2013.

Reggiani, L., Mattioli, E., Pittet, B., Duarte, L. V., Veiga de Oliveira, L. C., and Comas-Rengifo, M. J.: Pliensbachian (Early Jurassic) calcareous nannofossils from the Peniche section (Lusitanian Basin, Portugal): A clue for palaeoenvironmental reconstructions, Mar. Micropaleontol., 75, 1-16, 2010.

Reolid, M., Mattioli, E., Nieto, L. M., and Rodriguez-Tovar, F. J.: The Early Toarcian Oceanic Anoxic Event in the External Subbetic (Southiberian Palaeomargin, Westernmost Tethys): Geochemistry, nannofossils and ichnology, Palaeogeogr. Palaeoclimatol. Palaeoecol., 411, 79-94, http://dx.doi.org/10.1016/j.palaeo.2014.06.023, 2014.

Salaviale, C.: Climat, CCD et Nannofossiles Calcaires : Actions et Rétroactions au cours de l'Eocène moyen, Master Thesis, Université Claude Bernard Lyon1, 50 pp, 2013.

Shipboard Scientific Party: Initial Reports of the Deep Sea Drilling Project, edited by: de Graciansky, P. C., Poag, C. W., Cunningham, R. J., Loubere, P., Masson, D. G., Mazzullo, J. M., Montadert, L., Müller, C., Otsuka, K., Reynolds, L., Sigal, J., Snyder, S., Townsend, H. A., Vaos, S. P., and Waples, D., 1985.

Shipboard Scientific Party: Proceedings of the Ocean Drilling Program, Initial Reports, edited by: Erbacher, J., Mosher, D. C., Malone, M. J., Berti, D., Bice, K. L., Bostock, H., Brumsack, H.-J., Danelian, T., Forster, A., and Gladz, C., 2004.

Sprong, J., Youssef, M. A., Bornemann, A., Schulte, P., Steurbaut, E., Stassen, P., Kouwenhoven, T. J., and Speijer, R. P.: A multi-proxy record of the Latest Danian Event at Gebel Qreiya, Eastern Desert, Egypt, J. Micropalaeontol., 30, 167-182, http://dx.doi.org/10.1144/0262-821x10-023, 2011.

Stanley, S. M.: Effects of Global Seawater Chemistry on Biomineralization: Past, Present, and Future, Chem. Rev., 108, 4483-4498, 2008.

Stover, L. E., Brinkhuis, H., Damassa, S. P., de Verteuil, L., Helby, R. J., Monteil, E., Partridge, A. D., Powell, A. J., Riding, J. B., Smelror, M., and Williams, G. L.: Mesozoic-Tertiary dinoflagellates, acritarchs and prasinophytes, edited by: Jansonius, J., and McGregor, D. C., Palynology: Principles and Applications, American Association of Stratigraphic Palynologists Foundation, 2, 641–750, 1996.

Suan, G., Pittet, B., Bour, I., Mattioli, E., Duarte, L. V., and Mailliot, S.: Duration of the Early Toarcian carbon isotope excursion deduced from spectral analysis: Consequence for its possible causes, Earth Planet. Sci. Lett., 267, 666-679, 2008.

Suchéras-Marx, B., Giraud, F., Fernandez, V., Pittet, B., Lécuyer, C., Olivero, D., and Mattioli, E.: Duration of the Early Bajocian and the associated $\delta^{13}C$ positive excursion based on cyclostratigraphy, J. Geol. Soc. London, 170, 107-118, 2013.

Suchéras-Marx, B., Giraud, F., Mattioli, E., Gally, Y., Barbarin, N., and Beaufort, L.: Middle Jurassic coccolith fluxes: A novel approach by automated quantification, Mar. Micropaleontol., 111, 15-25, http://dx.doi.org/10.1016/j.marmicro.2014.06.002, 2014.

Suchéras-Marx, B., Guihou, A., Giraud, F., Lécuyer, C., Allemand, P., Pittet, B., and Mattioli, E.: Impact of the Middle Jurassic diversification of *Watznaueria* (coccolith-bearing algae) on the carbon cycle and $\delta^{13}C$ of bulk marine carbonates, Glob. Planet. Change, 86-87, 92-100, 2012.

Voigt, S., Erbacher, J., Mutterlose, J., Weiss, W., Westerhold, T., Wiese, F., Wilmsen, M., and Wonik, T.: The Cenomanian – Turonian of the Wunstorf section – (North Germany): global stratigraphic reference section and new orbital time scale for Oceanic Anoxic Event 2, Newsl. Stratigr., 43, 65-89, 2008.

Williams, J. R., and Bralower, T. J.: Nannofossil assemblages, fine fraction stable isotopes, and the palaeoceanography of the Valanginian-Barremian (Early Cretaceous) North Sea Basin, Paleoceanography, 10, 815-839, 1995.

Youssef Ali M (2009) High resolution calcareous nannofossil biostratigraphy and paleoecology across the Latest Danian Event (LDE) in central Eastern Desert, Egypt. *Mar Micropaleontol* 72:111–128.

Youssef Ali, M.: High resolution calcareous nannofossil biostratigraphy and paleoecology across the Latest Danian Event (LDE) in central Eastern Desert, Egypt, Mar. Micropaleontol., 72, 111-128, http://dx.doi.org/10.1016/j.marmicro.2009.03.007, 2009.

Zuzlewski, P.: Interactions entre production carbonatée (nannoplancton calcaire) et fluctuations de la CCD pendant le PETM (Paléocène–Eocène), Master Thesis, Université Claude Bernard Lyon1, 51 pp, 2014.